# IMPROVING THE EFFICIENCY OF CONFORMAL PREDICTORS VIA TEST-TIME AUGMENTATION

## ABSTRACT

In conformal classification, the goal is to output a *set* of predicted classes, accompanied by a probabilistic guarantee that the set includes the true class. Conformal approaches have gained widespread traction across domains because they can be composed with existing classifiers to generate predictions with probabilistically valid uncertainty estimates. In practice, however, the utility of conformal prediction is limited by its tendency to yield large prediction sets. We study this phenomenon and provide insights into why large set sizes persist, even for conformal methods designed to produce small sets. Using these insights, we propose a method to reduce prediction set size while maintaining coverage. We use test-time augmentation to replace a classifier's predicted probabilities with probabilites aggregated over a set of augmentations. Our approach is flexible, computationally efficient, and effective. It can be combined with any conformal score, requires no model retraining, and reduces prediction set sizes by up to 30%. We conduct an evaluation of the approach spanning three datasets, three models, two established conformal scoring methods, and multiple coverage values to show when and why test-time augmentation is a useful addition to the conformal pipeline.

## 1  INTRODUCTION

Machine learning classifiers excel at providing the most likely category for a particular input; where they fall short is in providing accurate notions of *uncertainty* for these predictions (Guo et al., 2017; Begoli et al., 2019; Kompa et al., 2021). Conformal prediction has emerged as a promising framework to provide existing classifiers with statistically valid uncertainty estimates by replacing the prediction of the most likely class with an *uncertainty set*, or a set of classes accompanied by a probabilistic guarantee that the true class appears in the set (Vladmir Vovk, 2005; Papadopoulos et al., 2007; Bates et al., 2021). Conformal prediction has been successfully applied to domains spanning clinical diagnosis (Papadopoulos et al., 2009; Lu et al., 2022a), drug discovery (Eklund et al., 2015; Alvarsson et al., 2021) and financial forecasting (Wisniewski et al., 2020).

Uncertainty sets are most useful when they achieve three desiderata: efficiency, adaptivity, and coverage. A conformal predictor that exhibits all three produces sets that are small, sized differently based on model certainty, and contain the true label at a pre-specified coverage rate (on average). Unfortunately, standard conformal prediction too often yields prediction sets that are uninformatively large (Babbar et al., 2022; Angelopoulos et al., 2022). For example, our experiments show that nearly every class in the iNaturalist 2021 dataset (Van Horn et al., 2021) receives prediction sets containing more than 100 species on average, using RAPS (Angelopoulos et al., 2022) (a conformal method designed to produce small set sizes) to achieve a coverage of 99%. This brings us to the goal of this work: to reduce prediction set sizes of standard conformal predictors, while maintaining adaptivity and coverage.

We accomplish this goal using ideas from test-time augmentation, a technique that has been used to improve the robustness and accuracy of conventional classifiers (Shanmugam et al., 2021; Cohen & Giryes, 2021; Perez et al., 2021; Lu et al., 2022b; Zhang et al., 2022; Enomoto et al., 2023). Test-time augmentation allows us to create an ensemble of predictions without training new models by simply perturbing the input with transformations. Our insight is to use test-time augmentation to improve the conformal score, a critical step in the conformal prediction pipeline. We learn the test-time augmentation policy using a small set of labeled data, which is commonly used by split

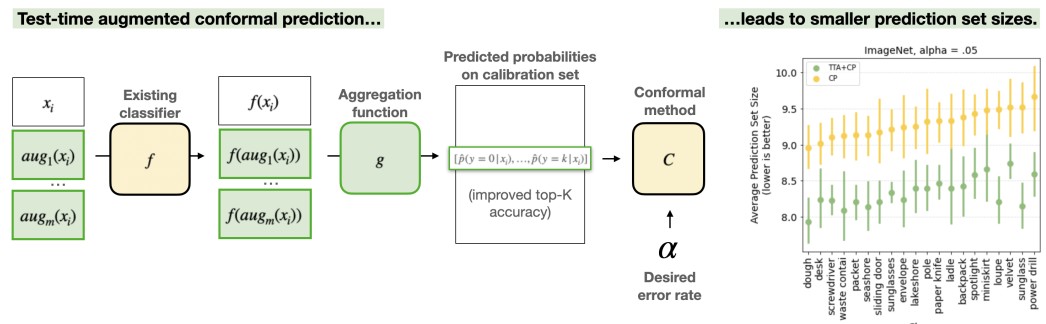

Figure 1: We illustrate the addition of test-time augmentation to conformal calibration in green (left) and provide a snapshot of the improvements it can confer (right). We show results on Imagenet, with a desired error rate of at most 5%, for the 20 classes with the largest predicted set sizes on average (computed over 10 calibration/test splits).

conformal predictors (the most efficient form of conformal prediction (Tibshirani et al., 2019)) to identify criteria for label inclusion in a prediction set.

Experiments across numerous datasets, base classifiers, and coverage specifications demonstrate the value of using test-time augmentation to improve the quality of the conformal score, which is directly tied to the efficiency of a conformal classifier. In fact, we show that our approach reduces set sizes for the classes with the *largest* prediction set sizes by up to 30%. We also show that the addition of test-time augmentation can bridge gaps between classifiers of different sizes; for example, test-time augmentation combined with ResNet-50 produces smaller set sizes than ResNet-101 alone.

**Central contribution.** The efficiency of conformal prediction hinges on the quality of the conformal score (Linusson et al., 2017). Together, our experiments show that one can easily improve the quality of a conformal score without retraining the model through the appropriate application of test-time augmentation. We provide an intuitive technique to reduce the prediction set sizes of existing conformal predictors using automatically learned test-time augmentations. We also provide a broad analysis of the relationship between model complexity, calibration set size, and conformal prediction performance. We view our work as a way to infuse domain-specific inductive biases into the conformal prediction framework and intend our work to broaden its practical utility.

## 2 RELATED WORK

Conformal prediction was first introduced in the 1990s by Gammerman et al. (1998); Saunders & Holloway (1999); Vladmir Vovk (2005) and in recent years, has become a popular approach to uncertainty quantification in machine learning (Barber et al., 2023). We review efforts to ensemble conformal predictors and efforts to reduce prediction set sizes below.

**Ensembles in conformal prediction** Several methods to generate ensembles of conformal predictors (e.g. cross-conformal prediction (Vovk, 2012), bootstrap conformal prediction (Vovk, 2015), aggregated conformal prediction (Carlsson et al., 2014; Linusson et al., 2017), and out-of-bag conformal prediction (Linusson et al., 2020)) are known to reduce set sizes and reduce the variance of the resulting conformal predictions, but all require training multiple base classifiers or conformal predictors. The approaches primarily differ in how data is sampled to create the training dataset for the classifier and the calibration set for the conformal predictor, and the estimated thresholds are typically averaged over the estimated conformal predictors. Our approach is distinct: we propose a technique to generate an ensemble from a *single* model by perturbing the input, which requires no additional base models and no additional conformal predictors.

**Efforts to improve efficiency in conformal prediction** In split conformal prediction, there are two ways to improve efficiency of a conformal predictor: adjustments to the conformal score or improvements to the underlying model. Many works have proposed new procedures to estimate and apply thresholds on conformal scores (Tibshirani et al., 2019; Bellotti, 2021; Angelopoulos et al.,

2022; Prinster et al., 2022; Ding et al., 2023). Romano et al. (2020) proposed APS, a conformal score based on the cumulative probability required to include the correct class in a prediction set. Angelopoulos et al. (2022) extended this work to propose RAPS, which modifies APS by penalizing the inclusion of low-probability classes. Concretely, RAPS does so by incrementing the predicted probabilities of the predicted probabilities of the $K - k_{reg}$ least likely classes by $\lambda$; the authors provide a procedure to automatically select $k_reg$ and $\lambda$ to minimize set size. Comparatively little work has focused on improvements to the underlying model. Jensen et al. (2022) ensemble a set of base classifiers to reduce set sizes, but generates each classifier by training models on subsets of the training data. Stutz et al. (2022) provide a way to train the base classifier and conformal wrapper jointly, instead of sequentially. In contrast, our work focuses on improving the underlying model *without* retraining, and can be easily combined with any of the above procedures; indeed, we see that the smallest prediction set sizes are achieved by the use of TTA *and* RAPS.

**Test-Time Augmentation**   Test-time augmentation (TTA) is a popular technique to improve the accuracy, robustness, and calibration of an existing classifier by aggregating predictions over a set of input transformations (Shanmugam et al., 2021; Perez et al., 2021; Zhang et al., 2022; Enomoto et al., 2023; Ayhan & Berens, 2018; Conde et al., 2023; Hekler et al., 2023). TTA has been successfully applied to a diverse range of predictive tasks including ICU survival modeling (Cohen et al., 2021), toxicity classification (Lu et al., 2022b), and plant identification (Igbineweka et al., 2020). Consequently, many have proposed new ways to perform TTA—for example, learning when to apply TTA (Mocerino et al., 2021), which augmentations to use (Kim et al., 2020; Lyzhov et al., 2020; Chun et al., 2022), or how to aggregate the resulting predictions (Shanmugam et al., 2021; Chun et al., 2022; Conde et al., 2023). While the value of test-time augmentation has been established in classification, the technique's impact in the context of conformal prediction is unknown. This forms the motivation for our work.

## 3   PROBLEM SETTING

We operate within the split conformal prediction framework, a widely used paradigm of conformal prediction. In this setting, a conformal classifier $\mathcal{C}(X_i) \subset \{1, \ldots, K\}$ maps input $X_i$ to a subset of $K$ possible classes and requires three inputs:

- Calibration set $D^{(cal)} = \{(X_1, Y_1), \ldots, (X_N, Y_N)\}$, containing $N$ labeled examples.
- Classifier $\hat{f} : \mathcal{X} \mapsto \Delta^K$, mapping inputs $X$ to a probability distribution over $K$ classes.
- Desired upper bound on error rate $\alpha \in [0, 1]$, where $(1 - \alpha)$ represents the probability the set contains the true class.

We study the introduction of two variables drawn from the test-time augmentation literature:

- Augmentation policy $\mathcal{A} = \{a_0, \ldots, a_m\}$, consisting of $m + 1$ augmentation functions, where $a_0$ is the identity transform. Policy $A(x_i)$ maps image $x_i$ to a set of inputs consisting of the original image and $m$ augmentations of the original image.
- Aggregation function $\hat{g}$, which aggregates a set of predictions to produce one prediction.

Each variable translates to a critical choice in test-time augmentation: what augmentations to apply ($\mathcal{A}$) and how to aggregate the resulting probabilities ($\hat{g}$). For fair comparison with approaches to conformal prediction, we evaluate the efficacy of TTA-augmented conformal prediction using metrics drawn from the conformal prediction literature: efficiency, coverage, and adaptivity (Angelopoulos & Bates, 2022). We provide exact definitions of each in Section 5. In plain language, the ideal conformal predictor produces small prediction sets (efficiency) that contain the true label at a rate greater than $(1 - \alpha)$ (coverage), where set size correlates with predictive uncertainty (adaptivity).

## 4   APPROACH

**Preliminaries**   Standard conformal prediction techniques yield prediction sets that are large and uninformative. Our goal is to learn – given an augmentation policy $\mathcal{A}$ – an aggregation function $\hat{g}$ to maximize the accuracy of the underlying classifier, and ultimately reduce the sizes of predict

sets generated from the classifier's predicted probabilities. We will briefly outline the conformal approach in this setting and then detail the mechanics of our method (illustrated in Figure 1); refer to Shafer & Vovk (2008) for a detailed introduction to conformal prediction.

Conformal predictors accept three inputs: a probabilistic classifier $f$, a calibration set $\mathcal{D}^{(cal)}$, and a pre-specified error rate $\alpha$. One can construct a conformal predictor from these inputs in three steps:

1. Define a score function $c(x, y)$, which produces a *conformal score* representing the uncertainty of the input example and label pair.
2. Produce a distribution of conformal scores across the calibration set by computing $c(x_i, y_i)$ for all $(x_i, y_i) \in \mathcal{D}^{(cal)}$.
3. Compute threshold $\hat{q}$ as the $\lceil (n+1)(1-\alpha) \rceil / n$ quantile of the distribution of conformal scores across all $n$ examples in the calibration set.

For a new example $x$, we compute $c(x, y)$ for all $y \in \{1, \ldots, K\}$, and include all $y_j$ for which $c(x, y_j) < \hat{q}$. We adopt the conformal score proposed by Romano et al. (2020), which equates to the *cumulative* probability required to include the correct class. Formally, this translates to:

$$\hat{\pi}_x(y') = \hat{p}(y = y'|x) = f(x)_{y'} \tag{1}$$

$$\rho_x(y) = \sum_{y'=1}^{K} \hat{\pi}_x(y') \mathbb{I}[\hat{\pi}_x(y') > \hat{\pi}_x(y)] \tag{2}$$

$$c(x, y) = \rho_x(y) + \hat{\pi}_x(y) \tag{3}$$

where $\hat{\pi}_x(y')$ corresponds to the predicted probability of class $y'$ given $x$, $\rho_x(y)$ is the cumulative probability of all classes predicted with higher probability than $y$. Conformal score $c(x, y)$ is thus composed of this cumulative probability and the predicted probability of class $y$. Consider an example input $x_i$ of a three-class problem where $y_i = 2$. If the predicted probabilities across classes 0, 1, and 2 were $[.5, .2, .3]$, the conformal score of the true class would be .8.

**Proposal** Our approach serves to update the conformal score by transforming the probabilities output by $f$ via test-time augmentation. Concretely, this replaces Equation 1 with the following, based on an augmentation policy $\mathcal{A}$ and aggregation function $g$.

$$\hat{\pi}_x(y') = \hat{p}(y = y'|x) = \hat{g}(f, x_i, \mathcal{A}) \tag{4}$$

We learn the aggregation function using the calibration set, $D^{(cal)}$. We define the aggregation function as an M-dimensional vector corresponding to augmentation-specific weights. We learn these weights using the calibration set by learning a set of weights that maximizes classifier accuracy by minimizing the cross-entropy loss computed between the predicted probabilities and true labels[1]. We parameterize $\hat{g}$ as follows:

$$\hat{g}(f, x_i, \mathcal{A}) = \Theta^T \mathbf{A}(f, x_i, \mathcal{A}) \tag{5}$$

where $\mathbf{A}$ uses $f$ to map input $x_i$ to a $M \times K$ matrix of predicted probabilities with $M$ being the number of augmentation transforms and $K$ being the number of classes. $\Theta$ is a $1 \times m$ vector corresponding to augmentation-specific weights. Each row in $\mathbf{A}(f, x_i, \mathcal{A})$ represents the pre-trained classifier's predicted probabilities on augmentation $a_m$ of $x_i$ or $f(a_m(x_i))$. In our experiments, TTA-Learned refers to TTA combined with augmentation weights learned by minimizing the cross entropy loss with respect to the true labels on the calibration set, while TTA-Avg refers to a simple average over the augmentations. We use distinct sets of examples to learn the test-time augmentation

---

[1]We found no significant difference between the use of cross-entropy loss and alternate losses considered in the conformal prediction literature (Table 3), and opt for the cross-entropy loss in favor of simplicity.

policy and to identify the threshold. Figure 8 shows that performance is not sensitive to the choice of $\beta$, so we use $\beta = .2$ in all experiments (please see Section A.10 for further discussion). While this does reduce the amount of data available to identify the appropriate threshold, we find that the benefits TTA confers *outweigh* the cost to threshold estimation. Importantly, we preserve the assumption of exchangeability, and therefore the coverage guarantee, because we apply the same augmentations $\mathcal{A}$ and aggregation function $\hat{g}$ to every example in the calibration set and all unseen examples, and we learn this aggregation using a completely distinct subset of e xamples. The computational cost scales with the size of $\mathcal{A}$; every additional augmentation translates to a forward pass of the base classifier. In principle, one could use the learned weights to save computation by identifying which test-time augmentations to generate.

## 5 EXPERIMENTAL SET-UP

We summarize details of our experimental set-up here, and will make code to reproduce all experiments publicly available.

**Datasets**   We show results on the test splits of three datasets: ImageNet (Deng et al., 2009) (50,000 natural images across 1,000 classes), iNaturalist (Van Horn et al., 2021) (100,000 images spanning 10,000 species), and CUB-Birds (Wah et al., 2011) (5,794 images representing 200 categories of birds). Images are distributed evenly over classes in ImageNet and iNaturalist, while CUB-Birds has between 11 and 30 images per class.

**Models**   The default model architecture, across all datasets, is ResNet-50 (He et al., 2016). The accuracies of our base classifiers are 76.1% (ImageNet), 76.4% (iNaturalist), and 80.5% (CUB-Birds). To characterize the relationship between model complexity and performance, we also provide results using ResNet-101 and ResNet-152 on ImageNet. For ImageNet, we make use of the pretrained models made available by PyTorch (Paszke et al., 2019). For iNaturalist, we use a model made public by (Niers, Tom, 2021). For CUB-Birds, we train our own network by finetuning the final layer of a ResNet-50 model initialized with ImageNet's pretrained weights.

**Augmentations**   We consider two augmentation policies. The first (the *simple* augmentation policy) consists of a random-crop and a horizontal-flip; to produce a random crop, we pad the original image with 4 pixels and take a 256x256 crop of the expanded image (thereby preserving the original image resolution). This augmentation policy is widely used because these augmentations are likely to be label-preserving. The second, which we will term the *expanded* TTA policy, consists of 12 augmentations: increase-sharpness, decrease-sharpness, autocontrast, invert, blur, posterize, shear, translate, color-jitter, random_crop, horizontal-flip, and random-rotation. An explanation of each augmentation is in the supplement (Sec. A.1). These augmentations are not always label preserving, but can improve performance when weights are learned, as in our work.

**Baselines**   We benchmark results using two conformal scores (translating to different definitions of $c(x, y)$ in Equation 3). The first score is APS (Romano et al., 2020) (described in Eqn. 3), which represents the cumulative probability required to include the correct class, and the second is RAPS (Angelopoulos et al., 2022), which modifies APS by adding a term to penalize large set sizes. For all experiments, we perform randomization of conformal scores during calibration and do not allow sets of size 0. We implement RAPS and APS using code provided by (Angelopoulos et al., 2022), and automatically select hyperparameters $k_{reg}$ and $\lambda$ to minimize set size. We also compare against conformal prediction using a simple average over the test-time augmentations (TTA-Avg). In the supplement, we also compare against non-conformal Top-1 and Top-5 prediction sets.

**Evaluation**   We evaluate results along the three axes commonly used in the conformal prediction literature: efficiency, coverage, and adaptivity. We quantify efficiency using two metrics: average prediction set size measured across all examples and class-average prediction set size, measured across all examples in a class. We define coverage as the percentage of outputted sets containing the true label. We draw our definition of adaptivity from the size-stratified coverage violation (SSCV) of (Angelopoulos et al., 2022). We first partition examples based upon the size of the prediction set for that example, using bins of $[0, 1], [2, 3], [4, 10],$ and $[101, ]$ set sizes. We then compute the average

coverage within each partition, and compute the maximum value of the theoretical coverage minus the actual coverage across partitions. The closer this value is to 0, the better the adaptivity.

For each dataset, we report results across 10 randomly generated splits into a calibration set and test set. For all experiments (save for the calibration set size experiment), the calibration set and test set are the same size. For the experiment studying calibration set size, we downsample the calibration set. We compute statistical significance using a paired t-test, with a Bonferroni correction (Weisstein, 2004) for multiple hypothesis testing.

| | | Expanded Aug Policy | | | Simple Aug Policy | | |
|---|---|---|---|---|---|---|---|
| Alpha | Method | ImageNet | iNaturalist | CUB-Birds | ImageNet | iNaturalist | CUB-Birds |
| 0.01 | RAPS | $37.751 \pm 2.334$ | $61.437 \pm 6.067$ | $\mathbf{15.293 \pm 2.071}$ | $37.751 \pm 2.334$ | $61.437 \pm 6.067$ | $\mathbf{15.293 \pm 2.071}$ |
| 0.01 | RAPS+TTA-Avg | $35.600 \pm 2.200$ | $57.073 \pm 5.914$ | $\mathbf{13.111 \pm 2.470}$ | $31.681 \pm 3.057$ | $\mathbf{54.169 \pm 6.319}$ | $14.550 \pm 1.425$ |
| 0.01 | RAPS+TTA-Learned | $\mathbf{31.248 \pm 2.177}$ | $\mathbf{53.195 \pm 4.884}$ | $14.045 \pm 1.323$ | $32.702 \pm 2.409$ | $\mathbf{51.391 \pm 5.211}$ | $13.803 \pm 1.734$ |
| 0.05 | RAPS | $5.637 \pm 0.357$ | $\mathbf{7.991 \pm 1.521}$ | $3.624 \pm 0.361$ | $5.637 \pm 0.357$ | $7.991 \pm 1.521$ | $3.624 \pm 0.361$ |
| 0.05 | RAPS+TTA-Avg | $5.318 \pm 0.113$ | $\mathbf{7.067 \pm 0.344}$ | $\mathbf{3.116 \pm 0.210}$ | $\mathbf{4.908 \pm 0.099}$ | $\mathbf{6.451 \pm 0.279}$ | $\mathbf{3.249 \pm 0.307}$ |
| 0.05 | RAPS+TTA-Learned | $\mathbf{4.889 \pm 0.168}$ | $\mathbf{6.682 \pm 0.447}$ | $3.571 \pm 0.576$ | $5.040 \pm 0.176$ | $\mathbf{6.788 \pm 0.496}$ | $\mathbf{3.290 \pm 0.186}$ |
| 0.10 | RAPS | $2.548 \pm 0.074$ | $2.914 \pm 0.116$ | $2.038 \pm 0.153$ | $2.548 \pm 0.074$ | $2.914 \pm 0.116$ | $2.038 \pm 0.153$ |
| 0.10 | RAPS+TTA-Avg | $2.470 \pm 0.071$ | $2.740 \pm 0.026$ | $\mathbf{1.780 \pm 0.139}$ | $2.327 \pm 0.086$ | $\mathbf{2.610 \pm 0.031}$ | $\mathbf{1.881 \pm 0.118}$ |
| 0.10 | RAPS+TTA-Learned | $\mathbf{2.312 \pm 0.054}$ | $\mathbf{2.625 \pm 0.043}$ | $1.893 \pm 0.187$ | $2.362 \pm 0.065$ | $2.638 \pm 0.026$ | $\mathbf{1.840 \pm 0.106}$ |

Table 1: Results across datasets for two augmentation policies and three coverage specifications. Each entry corresponds to the average prediction set size across 10 calibration/test splits. Bolded entries represent significantly better performance compared to the baseline (RAPS), or performance indistinguishable from the best approach. We report achieved coverage in Table 10.

## 6 RESULTS

We provide statistics on large prediction sets in Sec. 6.1 and present results on the improvements TTA confers across multiple datasets, coverage values, and augmentation policies in Sec. 6.2. We then dissect the dependence of these results on dataset and base classifier in Sec. 6.3 and the dependence on class in Sec. 6.4. We conclude with intuition behind why test-time augmentation improves the efficiency of conformal predictors in Sec. 6.5.

### 6.1 CURRENT CONFORMAL METHODS PRODUCE LARGE PREDICTION SET SIZES.

Here we characterize the extent to which large prediction sets occur and why they arise. Our results show that across datasets, conformal predictors often produce large prediction sets and that these sets consist of many low-probability classes.

For instance, let us consider the coverage (vs) prediction set size tradeoff made by RAPS, a widely used conformal prediction framework. For a coverage level of 99%, RAPS produces abnormally large prediction sets: 10% of examples receive a set size larger than 100 for Imagenet, 193 for iNaturalist, and 44 for CUB-Birds. Looking at the classes included in the prediction sets across all examples, we can see that a large percentage are associated with predicted probabilities lower than $1/(\# \text{ of classes})$: 47% for ImageNet, 62% for iNaturalist, and 45% for CUB-Birds. Even relaxing the coverage to 95% does not result in informative prediction sets: 10% of examples still receive large set sizes (ImageNet: $\geq 10$, iNaturalist: $\geq 14$, CUB-Birds: $\geq 6$).

Now consider an alternative, where each uncertainty set contains the top 10 predicted classes for each example: these sets achieve a coverage of 95.8% on ImageNet, 94.2% on iNaturalist, and 98.1% on CUB-Birds. The worst-case prediction set size in this setting is attractively small; sadly, this approach sacrifices the adaptivity and coverage guarantee that conformal predictors offer.

The existence of large prediction sets is not a criticism of RAPS; it corresponds to a limitation of underlying probabilistic classifier. There are two possible remedies: improvements to the conformal score, as many prior works have explored (Tibshirani et al., 2019; Angelopoulos & Bates, 2022; Guan, 2023), or improvements to the underlying classifier. As the next section will illustrate, test-time augmentation is a viable approach to improving the underlying classifier, and thereby the performance of conformal predictors.

## 6.2 TTA PRODUCES CONSISTENT AND SIGNIFICANT REDUCTIONS IN PREDICTION SET SIZE.

We compare against RAPS in the main text since it outperforms baselines in every comparison, and provide results comparing our method to APS and the Top-K baselines in the supplement (Sec. A.6 and Sec. A.5 respectively), along with replicates of each experiments across multiple $\alpha$ and datasets in Section A.8. We begin with results in the context of the expanded augmentation policy.

**Learned test-time augmentation policies**—RAPS+TTA-Learned in Table 1 and APS+TTA-Learned in Table 6)—**produce statistically and practically significant reductions in predicted set size**. TTA-Learned reduces prediction set sizes significantly in 16 of the 18 comparisons we conduct, and performs comparably in the remaining 2. Across all configurations —- different conformal scores, coverage guarantees, datasets, and models — the combination of RAPS, TTA-Learned, and the expanded augmentation policy, produce the smallest average set sizes.

When we compare learned augmentation weights (TTA-Learned) to a fixed average (TTA-Avg), we find that **TTA-Learned performs comparably or better than TTA-Avg.** in all comparisons using the expanded augmentation policy. This is intuitive: it is unlikely that all augmentations in the larger augmentation policy should be weighted equally. Indeed, when we look at the weights learned for the expanded augmentation policy, we see that several augmentations (blur, decrease sharpness, and shear) are consistently assigned a weight of 0, while certain augmentations are consistently included in learned policies (autocontrast, translate).

**While TTA improves both RAPS and APS, it produces improvements much larger in magnitude for APS** (up to 36% across datasets). This is because TTA serves to regularize the predicted probabilities in a way that is similar to (but mechanically different from) RAPS: it is depressing the maximum predicted probability and redistributing it over the remaining classes (by aggregating predictions that lie further away from the training distribution compared to the original input). This is why the expanded augmentation policy demonstrates such strong performance compared to the simple augmentation policy for APS: it translates to stronger regularization.

**TTA-Learned preserves coverage across all experiments**, since it respects the assumption of exchangeability, and in some cases, significantly improves coverage (exact values can be found in Tables 10 and 11). We next evaluate adaptivity via the size stratified coverage violation (SSCV). At low alpha ($\alpha = .01$, and $\alpha = .05$), the TTA-Learned's improvements to efficiency come at no cost to adaptivity. At higher alpha ($\alpha = .10$), there are three settings in which TTA-learned produces lower values for SSCV (significant according to a paired t-test). This suggests that while learned test-time augmentations may reduce adaptivity at higher $\alpha$ (e.g. .10), there is promising evidence that the improvements TTA-Learned confers at low $\alpha$ come at no cost to adaptivity.

## 6.3 DEPENDENCE ON DATASET, AUGMENTATION POLICY, AND BASE CLASSIFIER

**Dependence on dataset**  TTA consistently improves prediction set sizes on ImageNet and iNaturalist, but not CUB-Birds. This may be because the calibration set size for CUB-Birds (2,827 images) is an order of magnitude smaller than the calibration set for either ImageNet (25,000 images) and iNaturalist (50,000 images). Indeed, we find that TTA is more effective the larger the calibration set (Figure A.11).

**Dependence on augmentation policy**  We find that the expanded augmentation policy produces the larger reductions in set size compared to the simple augmentation policy. Although the introduction of many augmentations outside of the base classifier's train-time augmentation policy can decrease the top-1 accuracy of a classifier, both conformal scores considered in this work use the predicted probabilities for *all* classes. So while the expanded test-time augmentation policy may not result in a significantly more accurate classifier, it modifies the predicted probabilities for lower ranked classes. Larger augmentation policies also yield greater reductions in average prediction set size (Figure 3). That said, the simple augmentation policy does have its place: in the absence of a learned aggregation function, our results suggest the simple augmentation policy aggregated via an average can still improve the efficiency of conformal predictors (outperforming the original conformal score in 11 comparisons, matching performance in 3, and underperforming in 4).

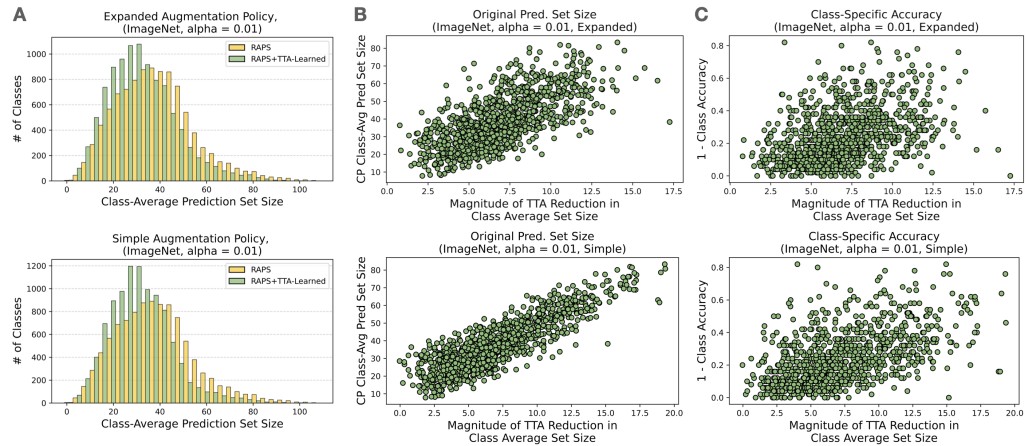

Figure 2: (A) **Class-average prediction set sizes.** Each point corresponds to the class-average prediction set size for a particular split. We plot results for ImageNet and $\alpha = .01$, where RAPS+TTA-Learned (green) produces a noticeable reduction in class-average prediction set sizes. (B, C) **Importance of original class-average prediction set size (middle) and class difficulty (right)**. Each point represents the mean class-average prediction set size across splits. TTA improvements are positively correlated with original class-average prediction set size (expanded: r = 0.67, $p <$ 1e-10) and class difficulty (expanded: r = 0.41, $p <$ 1e-10).

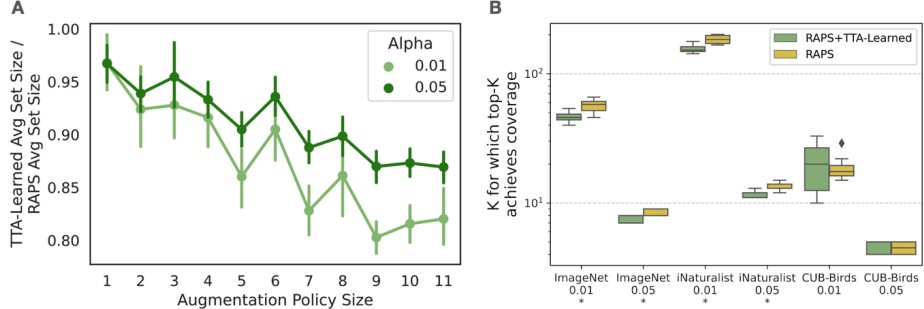

Figure 3: (A) **Size of augmentation policy**: We sample 5 policies of each size, with 10 calibration/test splits for each policy. Error bars correspond to 95 percentile intervals. Larger augmentation policies translate to consistent improvements in TTA's ability to reduce prediction set sizes. (B) **Optimal Top-K**: TTA-Learned significantly lowers the value of k required for Top-k prediction sets to achieve coverage on ImageNet and iNaturalist, but not CUB-Birds.

| | | Expanded Aug Policy | | | Simple Aug Policy | | |
|---|---|---|---|---|---|---|---|
| Alpha | Method | ResNet-50 | ResNet-101 | ResNet-152 | ResNet-50 | ResNet-101 | ResNet-152 |
| 0.01 | RAPS | 37.751 ± 2.334 | 33.624 ± 1.796 | 29.560 ± 3.481 | 37.751 ± 2.334 | 33.624 ± 1.796 | 29.560 ± 3.481 |
| 0.01 | RAPS+TTA-Avg | 35.600 ± 2.200 | 30.220 ± 1.774 | 27.203 ± 2.526 | **31.681 ± 3.057** | **27.206 ± 1.840** | **24.106 ± 2.100** |
| 0.01 | RAPS+TTA-Learned | **31.248 ± 2.177** | **25.722 ± 1.713** | **23.615 ± 1.656** | 32.702 ± 2.409 | 26.760 ± 1.974 | 24.765 ± 2.736 |
| 0.05 | RAPS | 5.637 ± 0.357 | 4.785 ± 0.102 | 4.376 ± 0.078 | 5.637 ± 0.357 | 4.785 ± 0.102 | 4.376 ± 0.078 |
| 0.05 | RAPS+TTA-Avg | 5.318 ± 0.113 | 4.433 ± 0.137 | 4.163 ± 0.185 | **4.908 ± 0.099** | **4.147 ± 0.122** | **3.868 ± 0.126** |
| 0.05 | RAPS+TTA-Learned | **4.889 ± 0.168** | **4.200 ± 0.200** | **3.824 ± 0.128** | 5.040 ± 0.176 | 4.194 ± 0.194 | 3.916 ± 0.356 |
| 0.10 | RAPS | 2.548 ± 0.074 | 2.267 ± 0.024 | 2.109 ± 0.027 | 2.548 ± 0.074 | 2.267 ± 0.024 | 2.109 ± 0.027 |
| 0.10 | RAPS+TTA-Avg | 2.470 ± 0.071 | 2.164 ± 0.031 | 2.049 ± 0.028 | 2.327 ± 0.086 | 2.093 ± 0.035 | **1.996 ± 0.018** |
| 0.10 | RAPS+TTA-Learned | **2.312 ± 0.054** | **2.099 ± 0.040** | **1.993 ± 0.026** | **2.362 ± 0.065** | **2.091 ± 0.041** | 1.988 ± 0.020 |

Table 2: Results across base classifiers for ImageNet. TTA-Learned can bridge the performance gap between different classifiers (for example, outperforming ResNet-152 alone when combined with ResNet-101), and yields significant reductions in set size regardless of the pretrained classifier used. We report achieved coverage in Table 11.

**Dependence on base classifier** We test the generalizability of our results to other models by rerunning the ImageNet experiments using ResNet-101 (accuracy of 77.4%) and ResNet-152 (accuracy of 78.3%). Unsurprisingly, more accurate models result in smaller prediction set sizes. TTA variants of conformal prediction again produce significant improvements in set size while maintaining coverage. We were particularly surprised to find that the combination of TTA with a smaller model — for example, ResNet-101 — can produce smaller set sizes than ResNet-152 alone. Concretely, when $\alpha$ is set to .01, RAPS+TTA-Learned and ResNet-101 produce set sizes that contain, on average, 26.5 classes, while RAPS and ResNet-152 produce an average set size of 29.6.

### 6.4 TTA IS MOST EFFECTIVE FOR CLASSES WITH THE LARGEST PREDICTION SETS

So far, we have established that TTA is a useful addition to the conformal pipeline on average. We now ask: where does this improvement come from, and what types of classes are responsible? We make two empirical observations. First, classes with larger predicted set sizes benefit most from the introduction of TTA; we plot this relationship in Figure 2, and find that a class's average prediction set size (computed over splits) is significantly associated with the change in set size TTA-Learned introduces (with the expanded augmentation policy and $\alpha = .01$, $r = 0.67$ and $p < 1e - 10$). Second, we find that class difficulty is significantly associated with changes in set size introduced by TTA (with the expanded augmentation policy and $\alpha = .01$, r = 0.41 and $p < 1e\text{-}10$), where prediction sets for difficult classes benefit more from TTA compared to their easier counterparts.

### 6.5 INTUITION

Why does the addition of test-time augmentation produce smaller prediction set sizes? In short, TTA improves top-K accuracy. We verify this claim by estimating $k$ such that the uncertainty sets comprised of the top $k$ predicted classes for each example achieve a coverage of $(1 - \alpha)$ on average. Indeed, we see that the probabilities updated by TTA — both with a simple average and learned weights — produce significantly lower values for $k$ compared to the originally outputted probabilities for both ImageNet and iNaturalist (Figure 3). This is *not* true for CUB-Birds, on which TTA offers little to no improvement. One could use such a procedure to determine whether TTA is worth adding to a conformal pipeline without collecting labeled data beyond the calibration set.

We can also understand the impact of TTA by studying the predicted probabilities associated with prediction set members. As observed earlier, large prediction sets are characterized by the inclusion of many low-probability classes. We measure the prevalence of these classes as the inclusion rate for classes with a predicted probability less than $1/(\text{# of classes})$; i.e. what percentage of prediction sets, on average, consists of classes which are less likely to be the true category than random chance. Considering ImageNet as an example, when we apply RAPS to generate prediction sets with 99% coverage, the addition of TTA reduces the mean and median percentage of low-probability classes in every split. These results suggest that TTA-adjusted probabilities are more informative, and that they address the symptom of large prediction sets—the inclusion of many low-probability classes—discussed in Section 6.1.

**Broader applications of TTA to conformal prediction** There are many other ways to combine test-time augmentation and conformal prediction. One might apply test-time augmentation during calibration (when computing conformal scores on the calibration set) and *not* during inference; this leads to smaller set sizes, but unsurprisingly breaks the coverage guarantee. The converse (TTA during inference and not calibration) maintains coverage but dramatically increases the prediction set sizes (because TTA depresses the maximum predicted probability, more classes can be included in the outputted set). Finally, one could consider the value of throwing away conformal prediction (and the guarantees it comes with) altogether, and creating a set out of the predictions made on each of the augmentations; refer to Section A.13 for further discussion.

## 7 DISCUSSION

We present a novel method for improving the efficiency of conformal predictors by using test-time augmentation to replace a classifier's predicted probabilities with probabilities aggregated over a set of transformations. Our results show that this technique is simple, efficient, and effective: it relies

on a labeled dataset already available to split conformal predictors, only requires multiple forward passes of a single classifier, and can reduce prediction set sizes by up to 30%. However, these results are not without limitations; we validate learned test-time augmentation policies in the context of image classification, and do not consider other modalities, for which appropriate transformations will substantially differ. It will also be important to characterize the value of learned test-time augmentation policies compared to more computationally expensive approaches to ensembling conformal predictors. In sum, our work charts a path towards practically useful conformal predictors by improving efficiency, without sacrificing adaptivity or coverage. More generally, it demonstrates yet another way in which test time augmentation can be easily combined with existing methods to improve their performance.

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

# A APPENDIX

## A.1 AUGMENTATIONS

The simple augmentation policy consists of a random crop and a horizontal flip, drawn from a widely used test-time augmentation policy in image classification Krizhevsky et al. (2012). The random crop pads the original image by 4 pixels and takes a 256x256 crop of the resulting image. The expanded augmentation consists of 12 augmentations; certain augmentations are stochastic, while others are deterministic. We design this set based on the augmentations included in AutoAugment (Cubuk et al., 2019). We exclude certain augmentations, however, to exclude 1) redundancies among augmentations and thereby make the learned weights interpretable and 2) augmentations are unlikely to be label-preserving. In particular, we exclude CutOut (because it is clearly not label-preserving in many domains) and exclude brightness, contrast, saturation, and color for their overlap with color-jitter. We also exclude contrast, because it is already modified via autocontrast, and equalize and solarize for their overlap with autocontrast and invert. This leaves us the following augmentations:

- *Shear*: Shear an image by some number of degrees, sampled between [-10, 10] (stochastic).
- *Translate*: Samples a vertical shift (by fraction of image height) from [0, .1] (stochastic).
- *Rotate*: Samples a rotation (by degrees) from [-10, 10] (stochastic).
- *Autocontrast*: Maximizes contrast of images by remapping pixel values such that the the lowest becomes black and the highest becomes white (deterministic).
- *Invert*: Inverts the colors of an image (deterministic).
- *Blur*: Applies Gaussian blur with kernel size 5 (and default $\sigma$ range of [.1, .2]) (stochastic).
- *Posterize*: Reduces the number of bits per channel to 4 (deterministic).
- *Color Jitter*: Randomly samples a brightness, contrast, and saturation adjustment parameter from the range [.9, 1.1] (stochastic).
- *Increase Sharpness*: Adjusts sharpness of image by a factor of 1.3 (deterministic).
- *Decrease Sharpness*: Adjusts sharpness of image by a factor of 0.7 (deterministic).
- *Random Crop*: Pads each image by 4 pixels, takes a 256x256 crop, and then proceeds to take a 224x224 center crop (stochastic).
- *Horizontal Flip*: Flips image horizontally (deterministic).

There are many possible expanded test-time augmentation policies; this particular policy serves as an illustrative example.

## A.2 LEARNING AGGREGATION FUNCTION $\hat{g}$

We learn $\hat{g}$ by minimizing the cross-entropy loss with respect to the true labels on the calibration set. Specifically, we learning the weights using SGD with a learning rate of .01, momentum of .9, and weight decay of 1e-4. We train each model for 50 epochs. There are natural improvements to this optimization, but this is not the focus of our work. Instead, our goal is to highlight the surprising effectiveness of TTA-Learned *without* the introduction of hyperparameter optimization.

## A.3 RESULTS OF COMPARISON TO TRAINING ON FOCAL LOSS

We expand Table 1 to include results for a variant of TTA-Learned which uses a focal loss in place of the cross-entropy loss. We conduct this exploration because empirically, the focal loss has been known to produce better-calibrated models. In practice, we see little difference between results when using a different loss function; RAPS+TTA-Leanred still outperforms RAPS + an average over the test-time augmentations, and RAPS alone. While this speaks to the method's flexibility to different loss functions, it is possible that the use of a loss function designed to reduce prediction set size could produce better performance.

## A.4 RESULTS OF COMPARISON TO DIFFERENT TEST-TIME AUGMENTATION WEIGHTING SCHEMES

One could weight each test-time augmentation by the accuracy achieved on the set of examples used to learn the test-time augmentation policy. We show results of doing so in Table 4. We in-

| | | Expanded Aug Policy | | Simple Aug Policy | |
|---|---|---|---|---|---|
| Alpha | Method | ImageNet | CUB-Birds | ImageNet | CUB-Birds |
| 0.01 | RAPS+TTA-Learned+Focal | $32.612 \pm 3.799$ | $13.416 \pm 1.991$ | $31.230 \pm 1.510$ | $15.503 \pm 2.364$ |
| 0.01 | RAPS+TTA-Learned+Conformal | $32.257 \pm 3.608$ | $13.776 \pm 2.198$ | $31.716 \pm 2.078$ | $14.432 \pm 2.184$ |
| 0.01 | RAPS+TTA-Learned+CE | $31.248 \pm 2.177$ | $14.045 \pm 1.323$ | $32.702 \pm 2.409$ | $13.803 \pm 1.734$ |
| 0.05 | RAPS+TTA-Learned+Focal | $4.906 \pm 0.195$ | $3.194 \pm 0.202$ | $4.956 \pm 0.239$ | $3.313 \pm 0.331$ |
| 0.05 | RAPS+TTA-Learned+Conformal | $4.867 \pm 0.122$ | $3.302 \pm 0.312$ | $4.996 \pm 0.405$ | $3.412 \pm 0.406$ |
| 0.05 | RAPS+TTA-Learned+CE | $4.889 \pm 0.168$ | $3.571 \pm 0.576$ | $5.040 \pm 0.176$ | $3.290 \pm 0.186$ |
| 0.10 | RAPS+TTA-Learned+Focal | $2.363 \pm 0.085$ | $1.791 \pm 0.102$ | $2.308 \pm 0.045$ | $1.860 \pm 0.131$ |
| 0.10 | RAPS+TTA-Learned+Conformal | $2.308 \pm 0.068$ | $1.865 \pm 0.163$ | $2.330 \pm 0.072$ | $1.868 \pm 0.122$ |
| 0.10 | RAPS+TTA-Learned+CE | $2.312 \pm 0.054$ | $1.893 \pm 0.187$ | $2.362 \pm 0.065$ | $1.840 \pm 0.106$ |

Table 3: Results across datasets for two augmentation policies and three coverage specifications using a focal loss. We set $\gamma$ to be 1, in line with prior work (Einbinder et al.). Each entry corresponds to the average prediction set size across 10 calibration/test splits. Both the focal and conformal loss do not outperform the cross-entropy loss; for simplicity, we report all results using the cross-entropy loss.

| | | Expanded Aug Policy | Simple Aug Policy |
|---|---|---|---|
| Alpha | Method | ImageNet | ImageNet |
| 0.01 | RAPS+TTA-Avg | $35.600 \pm 2.200$ | $\mathbf{31.681 \pm 3.057}$ |
| 0.01 | RAPS+TTA-Acc-Weighted | $37.115 \pm 4.112$ | $\mathbf{33.561 \pm 5.174}$ |
| 0.01 | RAPS+TTA-Err-Weighted | $36.012 \pm 3.501$ | $\mathbf{33.415 \pm 2.619}$ |
| 0.01 | RAPS+TTA-Learned | $\mathbf{31.723 \pm 1.737}$ | $32.702 \pm 2.409$ |
| 0.05 | RAPS+TTA-Avg | $5.318 \pm 0.113$ | $\mathbf{4.908 \pm 0.099}$ |
| 0.05 | RAPS+TTA-Acc-Weighted | $5.258 \pm 0.171$ | $\mathbf{4.942 \pm 0.242}$ |
| 0.05 | RAPS+TTA-Err-Weighted | $5.352 \pm 0.366$ | $\mathbf{4.859 \pm 0.139}$ |
| 0.05 | RAPS+TTA-Learned | $\mathbf{4.897 \pm 0.304}$ | $5.040 \pm 0.176$ |
| 0.10 | RAPS+TTA-Avg | $2.470 \pm 0.071$ | $\mathbf{2.327 \pm 0.086}$ |
| 0.10 | RAPS+TTA-Acc-Weighted | $2.443 \pm 0.068$ | $\mathbf{2.352 \pm 0.085}$ |
| 0.10 | RAPS+TTA-Err-Weighted | $2.416 \pm 0.076$ | $\mathbf{2.348 \pm 0.065}$ |
| 0.10 | RAPS+TTA-Learned | $\mathbf{2.290 \pm 0.064}$ | $\mathbf{2.362 \pm 0.065}$ |

Table 4: Results comparing learned weights to no augmentation-specific weights (TTA-Avg) and weights inferred from each test-time augmentation's accuracy (TTA-Acc-Weighted) or error (inverse weighting with respect to 1 - *aug_acc*). These results show that naive methods to weight the test-time augmentations can improve upon no learned weights at all, but learning the weights directly produces the best performance.

clude two variants of this approach: one in which each augmentations predictions are weighted by the classification accuracy of that augmented prediction (TTA-Acc-Weighted), and one in which each augmentation's predictions are inversely weighted with respect to the classification error on the labeled dataset. Unsurprisingly, this type of approach places too much weight on unhelpful augmentations. Learning the weights directly produces the best performance using the expanded augmentation policy. Learning the weights has little effect with the simple augmentation policy (a consistent result across all experiments).

## A.5 RESULTS OF COMPARISON TO TOP-1 AND TOP-5

We expand Table 1 to include the Top-1 and Top-5 baselines in Table 5. Unsurprisingly, neither outperform RAPS, and consequently none outperform the combination of RAPS, TTA-Learned, and the expanded augmentation policy.

## A.6 RESULTS USING APS

TTA-Learned combined with the expanded augmentation policy produces the smallest set sizes when combined with APS, across the datasets considered (Table 6) and each base classifier (Table 8). In contrast to the results using RAPS, TTA-Learned does not significantly outperform TTA-Avg when combined with APS. The central reason is that the improvements TTA confers — namely,

| | | ImageNet | | iNaturalist | | CUB-Birds | |
|---|---|---|---|---|---|---|---|
| Alpha | Method | Prediction Set Size | Empirical Coverage | Prediction Set Size | Empirical Coverage | Prediction Set Size | Empirical Coverage |
| 0.01 | Top-1 | 1.000 ± 0.000 | 0.761 ± 0.002 | 1.000 ± 0.000 | 0.766 ± 0.001 | 1.000 ± 0.000 | 0.804 ± 0.008 |
| 0.01 | Top-5 | 5.000 ± 0.000 | 0.928 ± 0.001 | 5.000 ± 0.000 | 0.915 ± 0.001 | 5.000 ± 0.000 | 0.959 ± 0.003 |
| 0.01 | RAPS | 37.751 ± 2.334 | 0.990 ± 0.001 | 61.437 ± 6.067 | 0.990 ± 0.001 | 15.293 ± 2.071 | 0.990 ± 0.001 |
| 0.01 | RAPS+TTA-Avg | 35.600 ± 2.200 | 0.991 ± 0.001 | 57.073 ± 5.914 | 0.990 ± 0.001 | 13.111 ± 2.470 | 0.991 ± 0.002 |
| 0.01 | RAPS+TTA-Learned | 31.248 ± 2.177 | 0.990 ± 0.001 | 53.195 ± 4.884 | 0.990 ± 0.001 | 14.045 ± 1.323 | 0.991 ± 0.002 |
| 0.05 | Top-1 | 1.000 ± 0.000 | 0.761 ± 0.002 | 1.000 ± 0.000 | 0.766 ± 0.001 | 1.000 ± 0.000 | 0.804 ± 0.008 |
| 0.05 | Top-5 | 5.000 ± 0.000 | 0.928 ± 0.001 | 5.000 ± 0.000 | 0.915 ± 0.001 | 5.000 ± 0.000 | 0.959 ± 0.003 |
| 0.05 | RAPS | 5.637 ± 0.357 | 0.951 ± 0.002 | 7.991 ± 1.521 | 0.954 ± 0.002 | 3.624 ± 0.361 | 0.955 ± 0.007 |
| 0.05 | RAPS+TTA-Avg | 5.318 ± 0.113 | 0.951 ± 0.001 | 7.067 ± 0.344 | 0.952 ± 0.002 | 3.116 ± 0.210 | 0.954 ± 0.007 |
| 0.05 | RAPS+TTA-Learned | 4.889 ± 0.168 | 0.952 ± 0.001 | 6.682 ± 0.447 | 0.954 ± 0.002 | 3.571 ± 0.576 | 0.957 ± 0.007 |
| 0.10 | Top-1 | 1.000 ± 0.000 | 0.761 ± 0.002 | 1.000 ± 0.000 | 0.766 ± 0.001 | 1.000 ± 0.000 | 0.804 ± 0.008 |
| 0.10 | Top-5 | 5.000 ± 0.000 | 0.928 ± 0.001 | 5.000 ± 0.000 | 0.915 ± 0.001 | 5.000 ± 0.000 | 0.959 ± 0.003 |
| 0.10 | RAPS | 2.548 ± 0.074 | 0.906 ± 0.004 | 2.914 ± 0.116 | 0.907 ± 0.003 | 2.038 ± 0.153 | 0.919 ± 0.014 |
| 0.10 | RAPS+TTA-Avg | 2.470 ± 0.071 | 0.905 ± 0.005 | 2.740 ± 0.026 | 0.908 ± 0.002 | 1.780 ± 0.139 | 0.912 ± 0.014 |
| 0.10 | RAPS+TTA-Learned | 2.312 ± 0.054 | 0.905 ± 0.004 | 2.625 ± 0.043 | 0.909 ± 0.003 | 1.893 ± 0.187 | 0.919 ± 0.016 |

Table 5: Results comparing performance against Top-K baselines. In each setting, conformal prediction produces either smaller set sizes, higher coverage, or both compared to the Top-K baselines.

improved top-k accuracy — do not address the underlying sensitivity of APS to classes with low predicted probabilities. As Angelopoulos et al. (2022) discuss, APS produces large prediction sets because of noisy estimates of small probabilities, which then end up included in the prediction sets. Both TTA-Learned and TTA-Avg smooth the probabilities: they reduce the number of low-probability classes by aggregating predictions over perturbations of the image. The benefit that both TTA-Learned and TTA-Avg add to APS is thus similar to how RAPS penalizes classes with low probabilities.

| | | Expanded Aug Policy | | | Simple Aug Policy | | |
|---|---|---|---|---|---|---|---|
| Alpha | Method | ImageNet | iNaturalist | CUB-Birds | ImageNet | iNaturalist | CUB-Birds |
| 0.01 | APS | 98.493 ± 3.075 | 131.681 ± 3.515 | 19.436 ± 0.995 | 98.493 ± 3.075 | 131.681 ± 3.515 | 19.436 ± 0.995 |
| 0.01 | APS+TTA-Avg | 68.714 ± 2.856 | **84.546 ± 3.655** | **17.715 ± 1.523** | 92.027 ± 4.797 | 145.401 ± 4.635 | 19.152 ± 1.667 |
| 0.01 | APS+TTA-Learned | 69.009 ± 2.156 | 85.093 ± 2.768 | 17.766 ± 1.608 | 90.613 ± 6.421 | 144.134 ± 4.371 | **18.552 ± 1.326** |
| 0.05 | APS | 19.820 ± 0.482 | 33.481 ± 0.786 | 5.921 ± 0.192 | 19.820 ± 0.482 | 33.481 ± 0.786 | 5.921 ± 0.192 |
| 0.05 | APS+TTA-Avg | 14.308 ± 0.279 | **26.021 ± 0.282** | **4.870 ± 0.208** | 18.862 ± 0.498 | 37.370 ± 0.735 | 6.306 ± 0.350 |
| 0.05 | APS+TTA-Learned | **14.084 ± 0.241** | 26.289 ± 0.529 | 4.913 ± 0.145 | 19.119 ± 0.479 | 36.940 ± 0.632 | 6.361 ± 0.480 |
| 0.10 | APS | 8.969 ± 0.158 | 16.755 ± 0.394 | 3.455 ± 0.164 | 8.969 ± 0.158 | **16.755 ± 0.394** | **3.455 ± 0.164** |
| 0.10 | APS+TTA-Avg | **7.193 ± 0.101** | **14.583 ± 0.333** | 3.108 ± 0.114 | 8.787 ± 0.136 | 18.300 ± 0.418 | 3.609 ± 0.135 |
| 0.10 | APS+TTA-Learned | 7.215 ± 0.106 | 14.538 ± 0.395 | **3.046 ± 0.073** | 8.813 ± 0.180 | 18.086 ± 0.420 | 3.638 ± 0.146 |

Table 6: We replicate our experiments across coverage levels and datasets using APS, another conformal score. TTA-Learned combined with the expanded augmentation policy produces the smallest set sizes across all comparisons. Interestingly, the simple augmentation policy is not as effective in the context of iNaturalist when using APS.

| | | Expanded Aug Policy | | | Simple Aug Policy | | |
|---|---|---|---|---|---|---|---|
| Alpha | Method | ImageNet | iNaturalist | CUB-Birds | ImageNet | iNaturalist | CUB-Birds |
| 0.01 | APS | 0.980 ± 0.001 | 0.986 ± 0.000 | 0.985 ± 0.001 | 0.980 ± 0.001 | **0.986 ± 0.000** | **0.985 ± 0.001** |
| 0.01 | APS+TTA-Avg | **0.985 ± 0.001** | **0.989 ± 0.001** | 0.989 ± 0.002 | **0.981 ± 0.001** | 0.987 ± 0.000 | **0.986 ± 0.003** |
| 0.01 | APS+TTA-Learned | **0.985 ± 0.001** | **0.989 ± 0.001** | **0.990 ± 0.002** | 0.980 ± 0.002 | 0.987 ± 0.000 | 0.985 ± 0.002 |
| 0.05 | APS | 0.931 ± 0.002 | 0.952 ± 0.001 | 0.945 ± 0.004 | 0.931 ± 0.002 | **0.952 ± 0.001** | **0.945 ± 0.004** |
| 0.05 | APS+TTA-Avg | **0.944 ± 0.002** | **0.956 ± 0.001** | 0.949 ± 0.005 | **0.937 ± 0.002** | 0.960 ± 0.001 | 0.949 ± 0.004 |
| 0.05 | APS+TTA-Learned | 0.943 ± 0.002 | **0.957 ± 0.001** | **0.950 ± 0.005** | **0.937 ± 0.002** | 0.959 ± 0.001 | 0.950 ± 0.005 |
| 0.10 | APS | 0.896 ± 0.002 | 0.923 ± 0.001 | 0.915 ± 0.006 | 0.896 ± 0.002 | **0.923 ± 0.001** | **0.915 ± 0.006** |
| 0.10 | APS+TTA-Avg | 0.903 ± 0.002 | **0.930 ± 0.001** | **0.920 ± 0.007** | **0.905 ± 0.002** | 0.933 ± 0.001 | **0.922 ± 0.005** |
| 0.10 | APS+TTA-Learned | **0.904 ± 0.002** | **0.930 ± 0.001** | 0.918 ± 0.006 | **0.906 ± 0.002** | 0.932 ± 0.001 | 0.922 ± 0.004 |

Table 7: Coverage values associated with experiments in Table 6. TTA-Learned produces significant improvements in coverage — larger in magnitude than in conjunction with RAPS — across when using the expanded augmentation policy. TTA-Learned produces no drops in coverage when using the simple augmentation policy, a nd produces improvements at $\alpha = .01$ and $\alpha = .05$.

| | | Expanded Aug Policy | | | Simple Aug Policy | | |
|---|---|---|---|---|---|---|---|
| Alpha | Method | ResNet-50 | ResNet-101 | ResNet-152 | ResNet-50 | ResNet-101 | ResNet-152 |
| 0.01 | APS | 98.493 ± 3.075 | 88.279 ± 4.121 | 79.231 ± 4.570 | 98.493 ± 3.075 | 88.279 ± 4.121 | 79.231 ± 4.570 |
| 0.01 | APS+TTA-Avg | **68.714 ± 2.856** | **64.197 ± 2.336** | **62.885 ± 3.125** | 92.027 ± 4.797 | 77.344 ± 2.214 | 73.377 ± 3.600 |
| 0.01 | APS+TTA-Learned | **69.009 ± 2.156** | **64.852 ± 2.823** | **64.045 ± 3.398** | 90.613 ± 6.421 | 78.627 ± 4.101 | 74.571 ± 3.516 |
| 0.05 | APS | 19.820 ± 0.482 | 15.830 ± 0.611 | 14.437 ± 0.591 | 19.820 ± 0.482 | 15.830 ± 0.611 | **14.437 ± 0.591** |
| 0.05 | APS+TTA-Avg | 14.308 ± 0.279 | **11.085 ± 0.267** | **10.605 ± 0.373** | 18.862 ± 0.498 | 15.039 ± 0.405 | 14.206 ± 0.499 |
| 0.05 | APS+TTA-Learned | **14.084 ± 0.241** | 11.118 ± 0.209 | 10.595 ± 0.368 | 19.119 ± 0.479 | 15.011 ± 0.346 | 14.252 ± 0.486 |
| 0.10 | APS | 8.969 ± 0.158 | 6.671 ± 0.175 | 6.134 ± 0.163 | 8.969 ± 0.158 | **6.671 ± 0.175** | **6.134 ± 0.163** |
| 0.10 | APS+TTA-Avg | **7.193 ± 0.101** | **5.454 ± 0.098** | **5.111 ± 0.096** | 8.787 ± 0.136 | 6.838 ± 0.143 | 6.309 ± 0.178 |
| 0.10 | APS+TTA-Learned | **7.215 ± 0.106** | **5.490 ± 0.090** | **5.131 ± 0.061** | 8.813 ± 0.180 | 6.826 ± 0.121 | 6.311 ± 0.123 |

Table 8: Results across base classifiers using APS alone, APS + TTA-Avg, and APS + TTA-learned in conjunction with the expanded augmentation policy (left) and simple augmentation policy (right). TTA-Learned and the expanded augmentation policy produce the smallest prediction sets (on average).

| | | Expanded Aug Policy | | | Simple Aug Policy | | |
|---|---|---|---|---|---|---|---|
| Alpha | Method | ResNet-50 | ResNet-101 | ResNet-152 | ResNet-50 | ResNet-101 | ResNet-152 |
| 0.01 | APS | 0.980 ± 0.001 | 0.979 ± 0.002 | 0.978 ± 0.002 | **0.980 ± 0.001** | **0.979 ± 0.002** | **0.978 ± 0.002** |
| 0.01 | APS+TTA-Avg | **0.985 ± 0.001** | **0.985 ± 0.001** | **0.984 ± 0.001** | 0.981 ± 0.001 | 0.980 ± 0.001 | 0.978 ± 0.002 |
| 0.01 | APS+TTA-Learned | **0.985 ± 0.001** | **0.985 ± 0.001** | **0.984 ± 0.001** | 0.980 ± 0.002 | 0.980 ± 0.002 | 0.979 ± 0.002 |
| 0.05 | APS | 0.931 ± 0.002 | 0.930 ± 0.002 | 0.929 ± 0.002 | 0.931 ± 0.002 | 0.930 ± 0.002 | 0.929 ± 0.002 |
| 0.05 | APS+TTA-Avg | **0.944 ± 0.002** | **0.942 ± 0.001** | **0.942 ± 0.002** | 0.937 ± 0.002 | 0.935 ± 0.002 | 0.934 ± 0.002 |
| 0.05 | APS+TTA-Learned | 0.943 ± 0.002 | **0.942 ± 0.001** | **0.942 ± 0.002** | 0.937 ± 0.002 | 0.935 ± 0.001 | 0.934 ± 0.002 |
| 0.10 | APS | 0.896 ± 0.002 | 0.892 ± 0.002 | 0.893 ± 0.002 | 0.896 ± 0.002 | 0.892 ± 0.002 | 0.893 ± 0.002 |
| 0.10 | APS+TTA-Avg | **0.903 ± 0.002** | **0.901 ± 0.001** | **0.902 ± 0.001** | 0.905 ± 0.002 | 0.903 ± 0.001 | 0.903 ± 0.002 |
| 0.10 | APS+TTA-Learned | **0.904 ± 0.002** | **0.902 ± 0.001** | **0.902 ± 0.001** | 0.906 ± 0.002 | 0.903 ± 0.002 | 0.903 ± 0.002 |

Table 9: Coverage values for APS and TTA variants of APS across base classifiers, using ImageNet. TTA-Learned or TTA-Avg in combination with the expanded augmentation policy significantly improve coverage in every comparison.

## A.7 RESULTS ON COVERAGE

| | | Expanded Aug Policy | | | Simple Aug Policy | | |
|---|---|---|---|---|---|---|---|
| Alpha | Method | ImageNet | iNaturalist | CUB-Birds | ImageNet | iNaturalist | CUB-Birds |
| 0.01 | RAPS | **0.990 ± 0.001** | **0.990 ± 0.001** | **0.990 ± 0.001** | **0.990 ± 0.001** | **0.990 ± 0.001** | **0.990 ± 0.001** |
| 0.01 | RAPS+TTA-Avg | **0.991 ± 0.001** | **0.990 ± 0.001** | **0.991 ± 0.002** | **0.990 ± 0.001** | **0.990 ± 0.001** | **0.991 ± 0.002** |
| 0.01 | RAPS+TTA-Learned | **0.990 ± 0.001** | **0.990 ± 0.001** | **0.991 ± 0.002** | **0.990 ± 0.001** | **0.990 ± 0.001** | **0.990 ± 0.002** |
| 0.05 | RAPS | **0.951 ± 0.002** | **0.954 ± 0.002** | **0.955 ± 0.007** | **0.951 ± 0.002** | **0.954 ± 0.002** | **0.955 ± 0.007** |
| 0.05 | RAPS+TTA-Avg | **0.951 ± 0.001** | **0.952 ± 0.002** | **0.954 ± 0.007** | **0.951 ± 0.001** | **0.953 ± 0.003** | **0.957 ± 0.004** |
| 0.05 | RAPS+TTA-Learned | **0.952 ± 0.001** | **0.954 ± 0.002** | **0.957 ± 0.007** | **0.951 ± 0.002** | **0.952 ± 0.002** | **0.956 ± 0.007** |
| 0.10 | RAPS | **0.906 ± 0.004** | **0.907 ± 0.003** | **0.919 ± 0.014** | **0.906 ± 0.004** | **0.907 ± 0.003** | **0.919 ± 0.014** |
| 0.10 | RAPS+TTA-Avg | **0.905 ± 0.005** | **0.908 ± 0.002** | **0.912 ± 0.014** | **0.905 ± 0.004** | **0.908 ± 0.002** | **0.915 ± 0.010** |
| 0.10 | RAPS+TTA-Learned | **0.905 ± 0.004** | **0.909 ± 0.003** | **0.919 ± 0.016** | **0.907 ± 0.004** | **0.908 ± 0.003** | **0.913 ± 0.011** |

Table 10: Coverage values for RAPS, RAPS+TTA-Avg, and RAPS+TTA-Learned across datasets and coverage values. RAPS+TTA-Learned never decreases the coverage achieved by RAPS alone, and in some cases, improves it significantly (as in the case of ImageNet and iNaturalist).

We provide exact values of coverage for each experiment here. In short, TTA-Learned combined with the expanded augmentation policy *never* worsens coverage, and in some cases, significantly improves it (although the improvements are small in magnitude). For those interested, we mirror each table describing average prediction set size with a table describing average coverage: coverage values for the RAPS experiment across coverage values and datasets can be found in Table 10 and coverage values for the RAPS experiment across base classifiers can be found in Table 11. Similarly, we provide coverage values for the APS experiment across datasets (Table 7) and across models (Table 8).

|  |  | Expanded Aug Policy | | | Simple Aug Policy | | |
|---|---|---|---|---|---|---|---|
| Alpha | Method | ResNet-50 | ResNet-101 | ResNet-152 | ResNet-50 | ResNet-101 | ResNet-152 |
| 0.01 | RAPS | **0.990 ± 0.001** | **0.990 ± 0.001** | **0.990 ± 0.001** | **0.990 ± 0.001** | **0.990 ± 0.001** | **0.990 ± 0.001** |
| 0.01 | RAPS+TTA-Avg | **0.991 ± 0.001** | **0.990 ± 0.001** | **0.990 ± 0.001** | **0.990 ± 0.001** | **0.990 ± 0.001** | **0.990 ± 0.001** |
| 0.01 | RAPS+TTA-Learned | **0.990 ± 0.001** | **0.990 ± 0.001** | **0.990 ± 0.001** | **0.990 ± 0.001** | **0.990 ± 0.001** | **0.990 ± 0.001** |
| 0.05 | RAPS | **0.951 ± 0.002** | **0.952 ± 0.002** | **0.952 ± 0.002** | **0.951 ± 0.002** | **0.952 ± 0.002** | **0.952 ± 0.002** |
| 0.05 | RAPS+TTA-Avg | **0.951 ± 0.001** | **0.951 ± 0.001** | **0.952 ± 0.002** | **0.951 ± 0.001** | **0.952 ± 0.002** | **0.952 ± 0.002** |
| 0.05 | RAPS+TTA-Learned | **0.952 ± 0.001** | **0.952 ± 0.002** | **0.952 ± 0.002** | **0.951 ± 0.002** | **0.952 ± 0.002** | **0.952 ± 0.002** |
| 0.10 | RAPS | **0.906 ± 0.004** | **0.906 ± 0.004** | 0.906 ± 0.002 | **0.906 ± 0.004** | 0.906 ± 0.004 | 0.906 ± 0.002 |
| 0.10 | RAPS+TTA-Avg | **0.905 ± 0.005** | 0.905 ± 0.002 | 0.908 ± 0.002 | **0.905 ± 0.004** | 0.908 ± 0.004 | 0.910 ± 0.002 |
| 0.10 | RAPS+TTA-Learned | **0.905 ± 0.004** | 0.907 ± 0.003 | 0.911 ± 0.002 | **0.907 ± 0.004** | 0.908 ± 0.004 | 0.910 ± 0.002 |

Table 11: Coverage values for TTA variants of conformal prediction compared to RAPS alone, across different base classifiers on ImageNet. TTA-Learned preserves coverage across all comparisons and significantly improves upon the achieved coverage using ResNet-101 with RAPS (granted, the magnitude of this improvement is small).

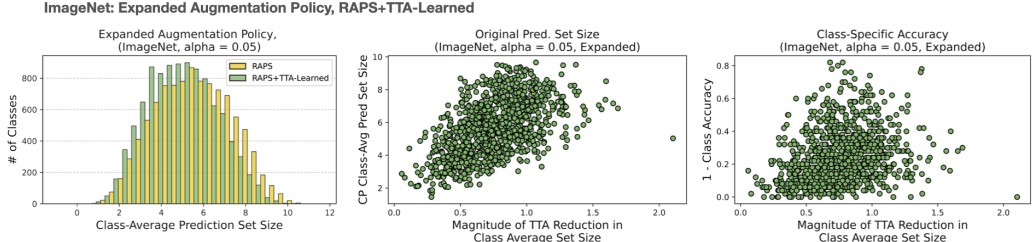

Figure 4: Class-specific performance for ImageNet, for a coverage of 95% $\alpha = .05$. Using the expanded augmentation policy RAPS+TTA-Learned produces a noticeable shift in class-average prediction set sizes to the left. There is a significant correlation between original prediction set size and improvements from TTA (middle) and between class difficulty and improvements from TTA (right).

### A.8 Replicated results with different alphas, datasets

We replicate the class-specific analysis for ImageNet at a value of $\alpha = .05$ (Figure 4), iNaturalist (Figure 5), and CUB-Birds (Figure 6). All trends are consistent with results in the main text, save for one notable exception: when TTA-Learned is applied to CUB-Birds, prediction set sizes of the classes with the *smallest* prediction set sizes and classes that are *easier* to predict benefit most from TTA. The significance of the relationship between original prediction set size and TTA improvement disappears when conducted on an example level in this setting. This could be a result of class imbalance in the dataset; it is possible that the class-average prediction set size obscures important variation in CUB-Birds.

### A.9 Impact of augmentation policy size

We also analyze the impact of augmentation policy size on average prediction set size for CUB-Birds (Figure 7), to understand if additional augmentations may produce larger reductions in set size than we observe. Larger augmentation policies appear to provide an improvement to average prediction set size at $\alpha = .05$, but offer little improvement for $\alpha = .01$.

### A.10 Impact of TTA data split

Learning the test-time augmentation policy requires a set of labeled data *distinct* from those used to select the conformal threshold. This introduces a trade-off: more labeled data for test-time augmentation may result in more accurate weights, but a less accurate conformal threshold, and vice versa. We study this tradeoff empirically in the context of ImageNet and the expanded augmentation policy and show results in Figure 8. We find that, as more data is taken away from the conformal calibration set, variance in performance grows. This is in line with our intuition; we have fewer ex-

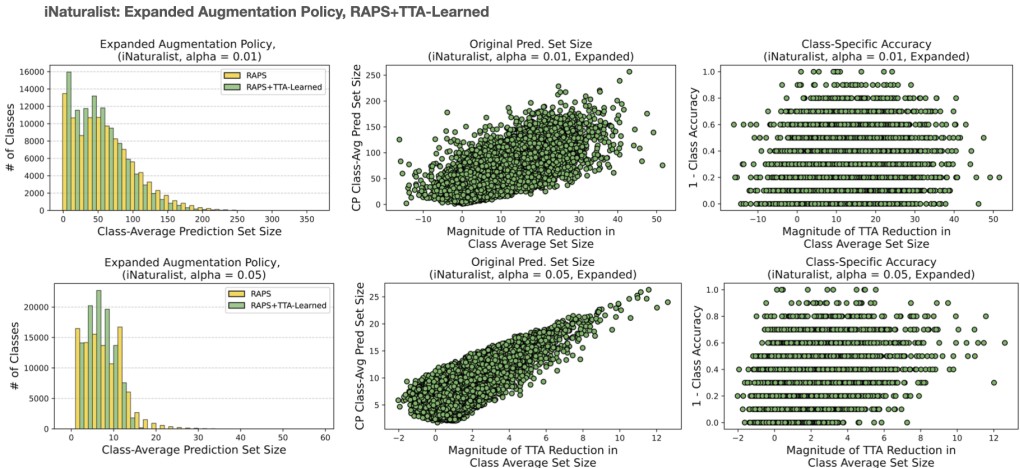

Figure 5: Class-specific performance for iNaturalist, for $\alpha = .01$ (top) and $\alpha = .05$ (bottom). We see a consistent relationship between TTA improvements and original class-average prediction set size (middle) and class difficulty (right). Estimates of class-specific accuracy on iNaturalist are quite noisy because there are 10 images per class (which produces distinct accuracy bands).

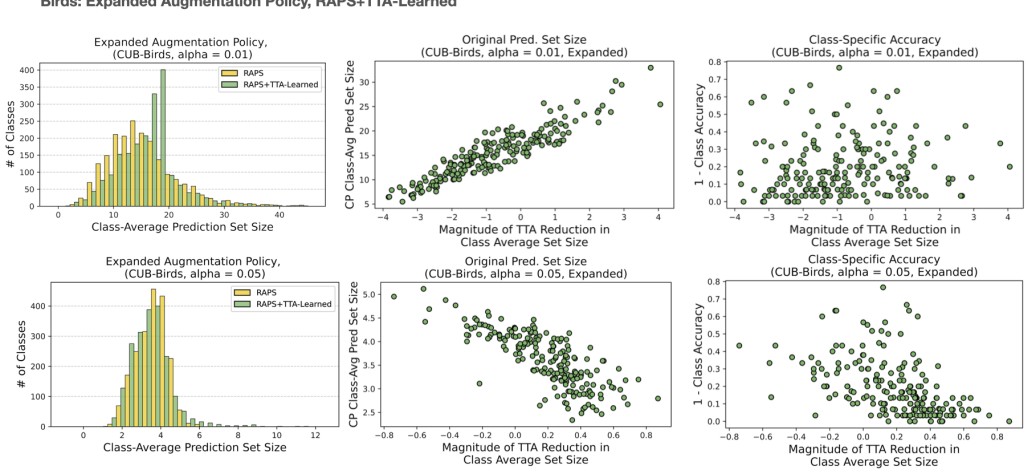

Figure 6: Class-specific performance for CUB-Birds, for $\alpha = .01$ (top) and $\alpha = .05\%$ (bottom). These graphs show an example for which TTA-Learned does *not* produce improvements in average prediction set size (computed across all examples). Interestingly, behavior on a class-specific level is different between $\alpha = .01$ and $\alpha = .05$. For $\alpha = .01$, results are consistent with other datasets: classes which originally receive large prediction set sizes and classes which are more difficult benefit most from the addition of TTA. For $\alpha = .05$, the exact opposite is true. While a majority of classes are hurt by TTA, classes that benefit from TTA are easier and receive smaller prediction set sizes.

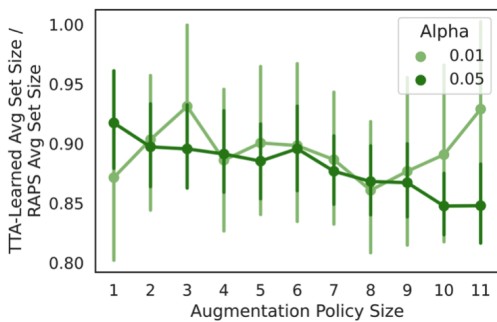

Figure 7: Impact of augmentation policy size on CUB-Birds. We see that larger policy sizes translate to a greater improvement (in terms of the ratio of average prediction set sizes using RAPS+TTA-Learned to average prediction set sizes using RAPS alone) for $\alpha = .05$. For $\alpha = .01$, there is no clear trend.

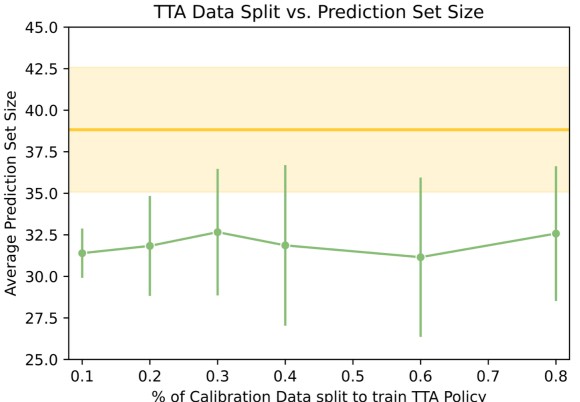

Figure 8: We plot the percentage of data used to train the TTA policy on the x-axis and the average prediction set size on the y-axis. Error bars describe variance over 10 random splits of the calibration and test set. We can make two observations: 1) as the data used to train the TTA policy increases and the data used to estimate the conformal threshold decreases, variance in performance grows and 2) across a wide range of data splits, learned TTA policies (green) introduce improvements to achieved prediction set sizes compared to the original probabilities (gold). These results also suggest that relatively little training data is required to learn a useful test-time augmentation policy; in this case, 2-3 images per class, or 10% of the available labeled data.

amples to approximate the distribution of conformal scores. However, at all percentages, test-time augmentation introduces a significant improvement in prediction set sizes over using all the labeled examples, and their original probabilities, to determine the threshold. This suggests that the benefits TTA confers outweigh the costs to the estimation of the conformal threshold, a practically useful insight to those who wish to apply conformal prediction in practice6

## A.11 IMPACT OF CALIBRATION SET SIZE

We plot the relationship between calibration set size and average prediction set size in Figure 9 across two augmentation policies, two datasets, and two values of $\alpha$. We see that TTA is more effective the larger the calibration set, in the context of ImageNet. In the context of CUB-Birds, it appears that TTA approaches equivalence with the conformal score alone as the calibration set size increases.

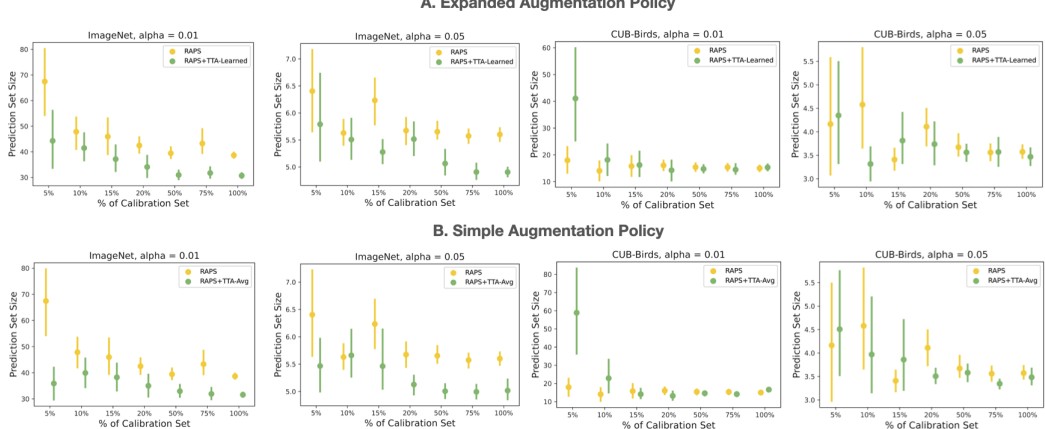

Figure 9: We plot the relationship between calibration set size and average prediction set size across two values of alpha, two augmentation policies, and two datasets (ImageNet and CUB-Birds). For ImageNet, larger calibration set sizes correlate with larger and more consistent improvements from the addition of TTA, where the improvement flattens out for calibration set sizes larger than 50%, or 12,500 images (12-13 per class). TTA does appear to be able to improve average prediction set size even with a calibration set size of 1,250 (5% of original ImageNet calibration set size). For CUB-Birds, a dataset on which TTA does not perform as well, we see that TTA performs comparably to RAPS alone the larger the calibration set.

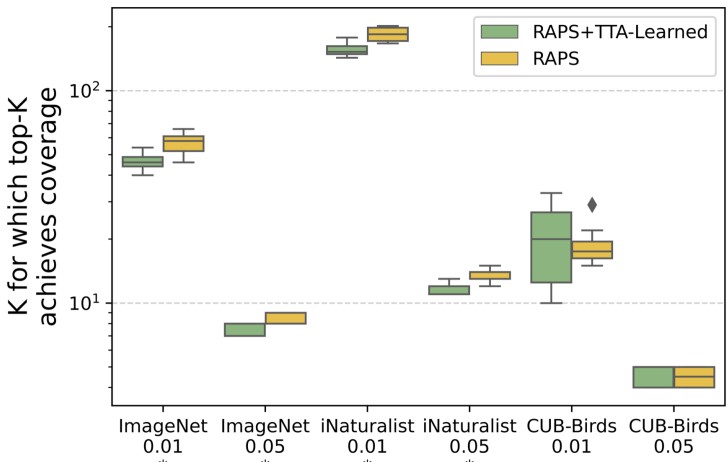

Figure 10: We plot the distribution of optimal $k$ for each dataset given two coverage values (.01 and .05). Probabilities transformed by TTA-Learned produce significantly lower values for $k$ (measured using a pairwise t-test) than the original probabilities on both ImageNet and iNaturalist, two datasets for which test-time augmentation produces consistent improvements.

## A.12    TTA'S EFFECT ON OPTIMAL TOP-$k$ FOR A GIVEN COVERAGE $\alpha$

As discussed in text, test-time augmentation improves the performance of conformal predictions by improving the top-k accuracy of the resulting probabilities, for some $k$. One way to understand this difference is to compare what value of $k_{opt}$ is necessary for a given coverage $\alpha$. Networks with higher top-k accuracy produce lower values of $k_{opt}$ than networks with low top-k accuracy. We visualize the difference in the optimal k for TTA-Learned probabilities compared to the original probabilities in Figure 10.

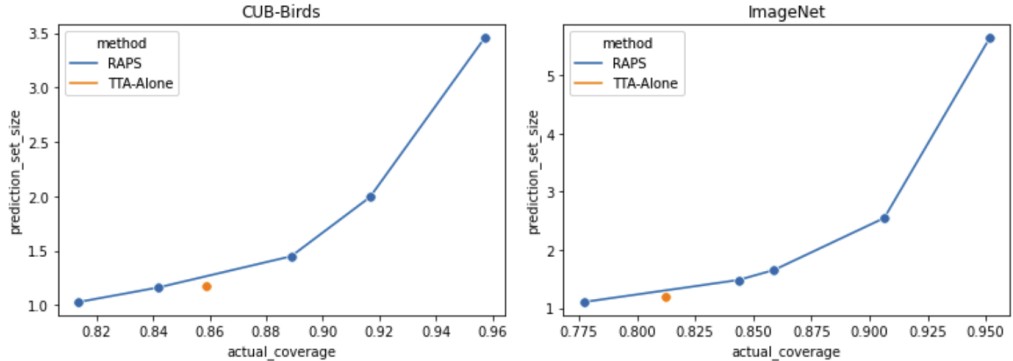

Figure 11: Comparison of uncertainty sets produced using the simple augmentation policy (orange) compared to the tradeoff RAPS achieves between prediction set size and coverage (blue).

## A.13 TTA UNCERTAINTY SETS

What if we instead generated uncertainty sets by creating a set out of the predictions made on each augmentations in a TTA policy? Interestingly, this approach can provide marginal improvements compared to the RAPS tradeoff between prediction set size and coverage—see Figure 11 for a comparison with the simple test-time augmentation policy. The sets are far less practically useful compared to those produced by a conformal predictor, but these differences may suggest ways to further improve the efficiency of conformal predictors.

