# OpenReview forum: "Improving the efficiency of conformal predictors via test-time augmentation"
_ICLR.cc/2024/Conference — Submitted to ICLR 2024_

### Official Review · Reviewer_L9GB · 2023-10-21

**Soundness:** 2 fair
**Presentation:** 1 poor
**Contribution:** 2 fair
**Rating:** 3
**Confidence:** 4

**Summary:**

The paper proposes a test-time augmentation methods to construct a stronger base model, which benefits conformal prediction in achieving better efficiency.

**Strengths:**

1) Efficiency of CP is an important problem.

2) Evaluation results look good.

**Weaknesses:**

(1) Lack of novelty: the paper mainly propose

(2) Lack of explanation: the idea is pretty simple but quite effective by looking at results, but the paper does not fully explain why this kind of method works so well. Simple test time augmentation does not bring so much improvement to me. I suggest providing top-k accuracy and adding more explanations on why it benefits conformal prediction so much accordingly.

(3) Lack of evaluation: I suggest adding comparisons to baselines of ensemble conformal prediction, mentioned in the related work. They also attempt to construct a stronger base model. It is essential to provide comparisons to them.

(4) Presentation: a) typo: $k_reg$ in the related work part; b) in section 3, indexing from 0 to N should induce N+1 samples; c) a large part in Section 4 is preliminary of conformal prediction, which should be introduced in Section 3 or a separate preliminary part.

**Questions:**

1) Can we analytically write out the optimal aggregation weights based on the empirical utility of each augmentation? (quite feasible to me) For example, augmentation with higher accuracy should have a larger aggregation weight. If that is the case, there is no need for optimization of those weights.

2) Why don't you parameterize the augmentations and also optimize the weights of augmentations?

3) Do you think about directly optimizing the efficiency objective (i.e., set size) as conformal training papers?

---

> ### Author Response · Authors · 2023-11-21
>
> Thanks for taking the time to share constructive, thoughtful feedback. We are glad to hear you believe the results are strong and the problem we focus on is important. Our responses to your suggestions are below:
>
> **Analytically determining augmentation weights**: We’ve included additional experiments comparing TTA-Learned to TTA-Acc-Weighted  (each augmentation is weighted by its accuracy) and TTA-Err-Weighted (each augmentation is weighted inversely to its classification error). An augmentation which achieves 70% accuracy compared to other augmentations (which each achieve, say, 90% accuracy) should not receive a non-zero weight, which it would under both TTA-Err-Weighted and TTA-Acc-Weighted. Exact results are included below for the expanded augmentation policy; Table 10 contains results for both the expanded and simple augmentation policy.
>
> | Dataset   |   alpha | Method                | Average Prediction Set Size   |
> |:----------|--------:|:----------------------|:------------------------------|
> | ImageNet  |    0.01 | RAPS+TTA-Acc-Weighted | 37.115 (4.112)                |
> | ImageNet  |    0.01 | RAPS+TTA-Err-Weighted | 36.012 (3.501)                |
> | ImageNet  |    0.01 | RAPS+TTA-Learned      | 31.248 (2.177)                |
> | ImageNet  |    0.05 | RAPS+TTA-Acc-Weighted | 5.258 (0.171)                 |
> | ImageNet  |    0.05 | RAPS+TTA-Err-Weighted | 5.352 (0.366)                 |
> | ImageNet  |    0.05 | RAPS+TTA-Learned      | 4.889 (0.168)                 |
> | ImageNet  |    0.1  | RAPS+TTA-Acc-Weighted | 2.443 (0.068)                 |
> | ImageNet  |    0.1  | RAPS+TTA-Err-Weighted | 2.416 (0.076)                 |
> | ImageNet  |    0.1  | RAPS+TTA-Learned      | 2.312 (0.054)                 |
>
> **On parametrizing the augmentations:** Thanks for the suggestion; this could certainly improve performance, but would also require more labeled data. Our results are meant to illustrate the surprising effectiveness of test-time augmentation given two relatively simple and general policies; what you suggest would be a good extension of our work.
>
> **Directly optimizing efficiency objective**: Good point; we ran two experiments to study this. In the first, we compare against the use of focal loss, since this loss is known to produce better calibrated classifiers, which correlates directly with smaller prediction set sizes. In the second, we compare against a conformal training loss, which directly optimizes for smaller sets. We found that neither produced significantly smaller prediction set sizes than using the cross-entropy loss. We’ve also included these results in the supplement of the updated manuscript (Table 3, reproduced below).
>
> | Dataset   |   alpha | Method                     | Average Prediction Set Size   |
> |:----------|--------:|:---------------------------|:------------------------------|
> | CUB-Birds |    0.01 | RAPS+TTA-Learned+CE        | 14.045 (1.323)                |
> | CUB-Birds |    0.01 | RAPS+TTA-Learned+Conformal | 13.776 (2.198)                |
> | CUB-Birds |    0.01 | RAPS+TTA-Learned+Focal     | 13.416 (1.991)                |
> | CUB-Birds |    0.05 | RAPS+TTA-Learned+CE        | 3.571 (0.576)                 |
> | CUB-Birds |    0.05 | RAPS+TTA-Learned+Conformal | 3.302 (0.312)                 |
> | CUB-Birds |    0.05 | RAPS+TTA-Learned+Focal     | 3.194 (0.202)                 |
> | CUB-Birds |    0.1  | RAPS+TTA-Learned+CE        | 1.893 (0.187)                 |
> | CUB-Birds |    0.1  | RAPS+TTA-Learned+Conformal | 1.865 (0.163)                 |
> | CUB-Birds |    0.1  | RAPS+TTA-Learned+Focal     | 1.791 (0.102)                 |
> | ImageNet  |    0.01 | RAPS+TTA-Learned+CE        | 31.248 (2.177)                |
> | ImageNet  |    0.01 | RAPS+TTA-Learned+Conformal | 32.257 (3.608)                |
> | ImageNet  |    0.01 | RAPS+TTA-Learned+Focal     | 32.612 (3.799)                |
> | ImageNet  |    0.05 | RAPS+TTA-Learned+CE        | 4.889 (0.168)                 |
> | ImageNet  |    0.05 | RAPS+TTA-Learned+Conformal | 4.867 (0.122)                 |
> | ImageNet  |    0.05 | RAPS+TTA-Learned+Focal     | 4.906 (0.195)                 |
> | ImageNet  |    0.1  | RAPS+TTA-Learned+CE        | 2.312 (0.054)                 |
> | ImageNet  |    0.1  | RAPS+TTA-Learned+Conformal | 2.308 (0.068)                 |
> | ImageNet  |    0.1  | RAPS+TTA-Learned+Focal     | 2.363 (0.085)                 |
>
>
> **Presentation**: Thanks for these edits, we’ve made changes to address them in the manuscript.
>
> **On improvements to top-k accuracy**: Thanks for the suggestion; we have included a comparison of the optimal k required to achieve coverage using the original probabilities, TTA-Average transformed probabilities, and TTA-Learned probabilities in Figure 10. TTA-Learned produces significantly lower values for the optimal k on the two datasets we see the largest improvements: ImageNet & iNaturalist.

---

### Official Review · Reviewer_KHLY · 2023-10-29

**Soundness:** 3 good
**Presentation:** 2 fair
**Contribution:** 1 poor
**Rating:** 3
**Confidence:** 4

**Summary:**

This paper introduces test-time augmentation (TTA) to significantly improve the efficiency of conformal predictors. In particular, by simply adapting TTA and learning a weight over augmentations, the produced augmentation-aggregated classifier provides a good scoring function that contributes to an efficient conformal predictor. The claim is empirically supported on the evaluation over three datasets and one baseline with various architectures.

**Strengths:**

This paper proposes a simple yet effective way to improve the efficiency of conformal predictors. In particular, this paper introduces TTA into the conformal prediction community.

**Weaknesses:**

I have a major concern on this paper. To my understanding, it is well-known that a better classifier (as a scoring function) provides smaller prediction sets in size (e.g., Table 1 of Angelopoulos et al., 2022 — I only chose papers with deep learning experiments); without these examples, it is clear that if we have a perfect classifier, the expected prediction set size is the smallest value (which is one).

Also, TTA with learnable parameters is at least firstly introduced in Shanmugam et al., 2021, which can be seen as making a better classifier from a base classifier by augmentations with learned weights over augmentations.

Given these, this paper revisited that a better classifier provides a more efficient prediction set, which is not new to me.


As an additional concern, this paper uses a calibration twice for learning weight parameters for augmentation in (5) and choosing a threshold for conformal prediction. This “double-dipping” should be avoided. I believe the results would not change too much but please use a calibration only once for choosing the threshold for conformal prediction.

**Questions:**

The following includes questions, which summarizes Weaknesses.

* It is not easy to accept the paper’s claim that this paper found a novel way to improve the efficiency of conformal prediction via TTA – it is well-known that a better classifier provides efficient conformal predictors. Also, a provided way of using TTA is not new. Please highlight novel points of this paper.
* For experiments, please conduct experiments by using a calibration set only once.

---

> ### Author Response · Authors · 2023-11-21
>
> Thanks for providing a thoughtful review. We believe this paper to be an important contribution to the literature on making conformal prediction practically useful, for the reason you describe: the method proposed is simple and effective.
>
> **On the novelty w.r.t. conformal prediction**: We agree that it is well known that better classifiers lead to better conformal predictors. We did not intend to convey the impression that this was a contribution of the paper. Instead, our goal is to demonstrate the surprising effectiveness of a simple and efficient method to improve the underlying model. Additionally, while there is literature on ensembles in conformal prediction, none present the possibility of a test-time ensemble, which is computationally cheaper than alternate approaches.
>
> **On the novelty w.r.t learned test-time augmentation policies**: We agree that the value of learned test-time augmentation policies has been established; however, a majority of the literature emphasizes the value of TTA to Top-1 classification accuracy. In contrast, our work shows that the benefits of test-time augmentation are not limited to the highest predicted probability; improvements in top-k accuracy produce significantly more efficient conformal predictors.
>
> **On using calibration set once**: This is a valid complaint. We’ve replicated all experiments by learning the TTA policy on a subset of the calibration set distinct from the subset used to identify the threshold. **There are no significant changes to the results**. We include an additional experiment in which we vary the percentage of data used to learn the test-time augmentation policy and find that results are largely not sensitive to the % of data used to train the TTA policy (Figure 8).

---

> > ### Comment · Reviewer_KHLY · 2023-11-23
> > **Thanks**
> >
> > Thanks for the response. As mentioned, I appreciate introducing the TTA in the context of conformal prediction. But, I believe that the value of conformal prediction is that it provides a handy-way to construct a prediction set for *any* given score functions. In other words, conformal prediction "helps" to identify the uncertainty of whatever given score functions -- if a score function is certain (e.g., a perfect classifier) on most examples, the prediction sets will be small; if the score function is uncertain (e.g., a random classifier) on most examples, the prediction sets will be large.
> >
> > Within the paper's context, two score functions are given: (1) a naive score function (=a less-certain score function) and (2) a TTAed-score function (=a more-certain score function). The paper simply shows that the more-certain score function has smaller prediction sets, which is a trivial consequence of conformal prediction.
> >
> > If this paper has value as a scientific paper, we can produce many scientific papers by simply replacing the score function into a better one (e.g., whenever the community discovers better architectures), applying conformal prediction, and demonstrating the efficiency of the better score function, which looks not desirable.
> >
> > Finally, for "our goal is to demonstrate the surprising effectiveness of a simple and efficient method to improve the underlying model", it is already demonstrated by the TTA paper.
> >
> > For these reasons, I'll maintain my score, but let me know if I misunderstood anything.

---

> > > ### Author Response · Authors · 2023-11-23
> > > **thanks for your response!**
> > >
> > > Thanks for continuing the discussion and engaging with our response.
> > >
> > > > The paper simply shows that the more-certain score function has smaller prediction sets, which is a trivial consequence of conformal prediction.
> > >
> > > We would say we go beyond this and can clarify what we mean. TTA-Learned is distinct from other methods which produce more-certain score functions (e.g. training a fancy new architecture or collecting additional data) because it requires only the ingredients available to split conformal predictors (i.e. a labeled calibration set, a pre-trained model, and a score function). Despite its simplicity, we show that TTA-Learned results in significant reductions in set size across datasets, models, and coverage guarantees. This was previously unknown, and is of practical importance to those who wish to apply conformal prediction.
> > >
> > > Our results also speak to the tradeoff between using labeled data to improve the conformal score (as with TTA-Learned) and using labeled data to identify the conformal threshold. We show that you can achieve much better efficiency by using a portion of the labeled data to improve the conformal score, instead of reserving all examples to identify the conformal threshold. We see our work as a step towards thinking about the question of how to best use labeled data in conformal prediction, which we think will be key to real-world adoption. We appreciate the discussion, which has convinced us to make this point more clear in the manuscript.
> > >
> > > > Finally, for "our goal is to demonstrate the surprising effectiveness of a simple and efficient method to improve the underlying model", it is already demonstrated by the TTA paper.
> > >
> > > The TTA paper you reference demonstrates the effectiveness of learned TTA policies for improving Top-1 image classification accuracy of trained models. For example, they show that learned test-time augmentation policies introduce an improvement of ~1% in classification accuracy for ResNet-50+ImageNet. For the same combination, we show a 16% reduction in prediction set size at a coverage guarantee of 99%. While the works share a focus on learned test-time augmentation policies, our results do not follow directly from theirs.
> > >
> > > The insight in our work is that TTA changes much more than the highest predicted probability; indeed, it significantly improves the optimal k at which Top-k sets achieve coverage (Figure 10 in the revised manuscript). In short, our findings on the relationship between TTA and conformal predictors are distinct from prior lines of work and their focus on the maximum predicted probability.
> > >
> > > Thanks once again for reading our paper & rebuttal closely; we hope this clarifies where we see our work falling within the literature on conformal prediction.

---

### Official Review · Reviewer_Gekq · 2023-10-29

**Soundness:** 3 good
**Presentation:** 4 excellent
**Contribution:** 3 good
**Rating:** 6
**Confidence:** 4

**Summary:**

The paper describes a data-augmentation approach to improve the efficiency of CP prediction sets.  Instead of evaluating a single model prediction, the conformity measure depends on a set of predictions through a trainable aggregation function. The authors show empirically that training the aggregation function through a cross-entropy loss improves the efficiency of the resulting prediction intervals.

**Strengths:**

The idea is simple but looks powerful. The amount of empirical evidence provided is notable.

**Weaknesses:**

The authors should clarify why their idea is different from replacing the underlying model with an ensemble method. The difference would be clear if the aggregation weights were trained by optimizing the CP efficiency directly. But the learning strategy is "minimizing the cross entropy loss with respect to the true labels on the calibration set". The link between the cross-entropy loss and the size of the prediction sets is not explicit.

**Questions:**

- An ablation study is run to compare different underlying models. It would be interesting to see what happens if the underlying model is an ensemble method, e.g. a random forest algorithm.
- The aggregation function is trained by "minimizing the cross entropy loss with respect to the true labels on the calibration set". Does this preserve the marginal validity of the prediction sets?
- Have you compared with any adaptive CP approaches like [1]?

[1]  Romano, Yaniv, Matteo Sesia, and Emmanuel Candes. "Classification with valid and adaptive coverage." Advances in Neural Information Processing Systems 33 (2020): 3581-3591.

---

> ### Author Response · Authors · 2023-11-21
>
> Thanks for your positive and constructive review. We agree that the proposed method is both simple and powerful, with strong empirical evidence that it produces consistent improvements over a variety of datasets. We also agree that the paper would be improved by addressing the issues you raise.
>
> > The authors should clarify why their idea is different from replacing the underlying model with an ensemble method.
>
> Thanks for this suggestion. In short, our method is different because it does not require multiple models, or the computational costs associated with training multiple models. This is what makes test-time augmentation an attractive approach in image classification; in our work, we show that it is a wise choice in conformal prediction.
>
> > The difference would be clear if the aggregation weights were trained by optimizing the CP efficiency directly. But the learning strategy is "minimizing the cross entropy loss with respect to the true labels on the calibration set".
>
> Your point is valid. We have added experiments that learn the TTA policy by minimizing two alternate losses: the focal loss and a conformal training loss, both of which have more direct ties to small set sizes. We found that neither loss significantly improved upon using cross-entropy loss (Table 3, reproduced below for convenience).
>
> | Dataset   |   alpha | Method                     | Average Prediction Set Size   |
> |:----------|--------:|:---------------------------|:------------------------------|
> | CUB-Birds |    0.01 | RAPS+TTA-Learned+CE        | 14.045 (1.323)                |
> | CUB-Birds |    0.01 | RAPS+TTA-Learned+Conformal | 13.776 (2.198)                |
> | CUB-Birds |    0.01 | RAPS+TTA-Learned+Focal     | 13.416 (1.991)                |
> | CUB-Birds |    0.05 | RAPS+TTA-Learned+CE        | 3.571 (0.576)                 |
> | CUB-Birds |    0.05 | RAPS+TTA-Learned+Conformal | 3.302 (0.312)                 |
> | CUB-Birds |    0.05 | RAPS+TTA-Learned+Focal     | 3.194 (0.202)                 |
> | CUB-Birds |    0.1  | RAPS+TTA-Learned+CE        | 1.893 (0.187)                 |
> | CUB-Birds |    0.1  | RAPS+TTA-Learned+Conformal | 1.865 (0.163)                 |
> | CUB-Birds |    0.1  | RAPS+TTA-Learned+Focal     | 1.791 (0.102)                 |
> | ImageNet  |    0.01 | RAPS+TTA-Learned+CE        | 31.248 (2.177)                |
> | ImageNet  |    0.01 | RAPS+TTA-Learned+Conformal | 32.257 (3.608)                |
> | ImageNet  |    0.01 | RAPS+TTA-Learned+Focal     | 32.612 (3.799)                |
> | ImageNet  |    0.05 | RAPS+TTA-Learned+CE        | 4.889 (0.168)                 |
> | ImageNet  |    0.05 | RAPS+TTA-Learned+Conformal | 4.867 (0.122)                 |
> | ImageNet  |    0.05 | RAPS+TTA-Learned+Focal     | 4.906 (0.195)                 |
> | ImageNet  |    0.1  | RAPS+TTA-Learned+CE        | 2.312 (0.054)                 |
> | ImageNet  |    0.1  | RAPS+TTA-Learned+Conformal | 2.308 (0.068)                 |
> | ImageNet  |    0.1  | RAPS+TTA-Learned+Focal     | 2.363 (0.085)                 |
>
> > The aggregation function is trained by "minimizing the cross entropy loss with respect to the true labels on the calibration set". Does this preserve the marginal validity of the prediction sets?
>
> Excellent point; thank you for bringing it up. We have replicated all experiments by learning the TTA policy on a distinct set of examples from those used to identify the conformal threshold and find no significant changes to results. We've updated the methods section to reflect this data split, and include an additional experiment showing that TTA's  results are robust to any particular choice of data split.
>
> > Have you compared with any adaptive CP approaches like [1]?
>
> We compare to [1] in Table 6 (APS) and work that builds on it in Table 1 (RAPS). We have made this more clear in the updated manuscript.

---

### Official Review · Reviewer_YZ5v · 2023-10-30

**Soundness:** 3 good
**Presentation:** 3 good
**Contribution:** 3 good
**Rating:** 6
**Confidence:** 3

**Summary:**

The paper addresses the challenge in conformal classification where it often produces excessively large prediction sets. To tackle this, the authors introduce an approach leveraging test-time augmentation (TTA). This method replaces a classifier's predicted probabilities with those aggregated over various augmentations. Notably, this approach is flexible, doesn't require model retraining, and has shown to reduce prediction set sizes by up to 30%. The paper's robust experimental evaluation spans multiple datasets, models, and conformal scoring methods, underscoring the effectiveness and applicability of their TTA-based solution.

**Strengths:**

Strengths:

The paper proposes an interesting approach to the existing challenge in conformal classification of producing large prediction sets. The idea of utilizing test-time augmentation (TTA) to address this is both innovative and timely.

The approach is model-agnostic, which makes it potentially widely applicable.

The paper provides insights into the conformal classification's tendency to yield large prediction sets, which can deepen understanding in the area.

The evaluation spans multiple datasets, models, and conformal scoring methods, suggesting a thorough empirical investigation.

**Weaknesses:**

While the paper preserves the assumption of exchangeability, it would be helpful to discuss any potential impacts or corner cases where this might not hold true.

How does the addition of test-time augmentation impact the computational efficiency of predictions, especially in real-time applications?

The paper claims the approach is flexible. However, is there a range or type of augmentation that works best for certain kinds of datasets or problems?

**Questions:**

How did the authors decide on the specific augmentations for the test-time augmentation? A more detailed breakdown would help the reader understand the decision-making process.

Could the authors provide more real-world scenarios or case studies where their approach would be particularly beneficial?

It would be helpful if the authors could discuss any potential limitations of their method, and how they might be addressed in future iterations or research.

While the paper provides an evaluation of the proposed approach, a more direct comparison with other recent methods aiming to reduce prediction set sizes in conformal classification would be beneficial.

---

> ### Author Response · Authors · 2023-11-21
>
> Thanks for spending the time to write a constructive and detailed review; we are glad to hear that you think the approach is interesting, that its model-agnosticity is a notable strength, and that you believe the experimentation to be thorough.
>
> **On the choice of augmentation policy**: We drew our augmentation policies from the test-time augmentation literature; one is inspired by the classic augmentation policy used to boost image classification accuracy (flips and crops), and the other is drawn from prior work on learning train-time augmentation policies [1]. We have made these connections more clear in the updated manuscript. Our results show that relatively simple and general test-time augmentation policies can produce large improvements to conformal predictors. That isn’t to say that other augmentations might not prove a fruitful direction for future work.
>
> **On computational efficiency**: good point. Inference scales linearly with the number of augmentations. One can mitigate this cost through parallelism, or by not generating augmentations with an assigned weight of 0. This is covered in the revised manuscript.
>
> **On real-world scenarios**: We believe that small set size is especially critical in healthcare, where the human bandwidth to consider a large number of options is limited. We are working to incorporate an experiment on a dermatology classification dataset.
>
> **On potential limitations**: Our proposed method has at least three limitations. The first is, as you describe, computational inefficiency. The second is the dependence on using labeled data to train the TTA policy, which reduces the data available to estimate the threshold. There is a tradeoff between the two, but our empirical results suggest that there is a large region of data splits for which TTA produces improvements. The third is the restricted search space of transformations; one can imagine stronger results with a more flexible space of transforms, but this would also likely require more labeled data.
>
> [1] Cubuk, Ekin D., et al. "Autoaugment: Learning augmentation strategies from data." Proceedings of the IEEE/CVF conference on computer vision and pattern recognition. 2019.

---

> > ### Comment · Reviewer_YZ5v · 2023-11-22
> >
> > I thank the authors for the clarification. I maintain my favorable score for the paper.

---

### Official Review · Reviewer_aF3F · 2023-10-31

**Soundness:** 1 poor
**Presentation:** 3 good
**Contribution:** 2 fair
**Rating:** 5
**Confidence:** 5

**Summary:**

The proposed approach is conceptually simple and uses test time augumentation to improve the predictions of the underlying classifiers which in turn imporves the conformal scores which then results in smaller (more efficient) prediction sets. This has the benefit that the model does not need to be retrained thus maintaining the flexibility of CP and increasing applicability. The authors compare 4 variants based on learned vs. simple average of the augmentations, and simple vs. expanded policy. Since this approach is orthogonal to any improvements in the score function it can be applied on top of different scores (e.g. RAPS or APS).

The authors also investigate some of the reasons behind the improvement and identify that e.g. one reason is due to the the improved top-$k$ accuracy. The evaluation w.r.t. datasets such as iNaturalist that contain an order of magnitude more classes than ImageNet is appreciated.

**Strengths:**

I think the simplicity of the approach is its biggest stregth. It is easy to implement, easy to understand, and it still seems to yield consistent improvements across different settings and datasets.

The experimental evaluation is reasonably detailed and the ablation analysis (e.g. simple vs. expanded, learned vs. average) is informative.

**Weaknesses:**

The biggest weakness of the approach is that the conformal guarantee is broken, or at least it has not been formally proved to hold for the learned setting.

The author state that "We learn these weights using the calibration set by learning a set of weights that maximizes classifier accuracy by minimizing the cross-entropy loss computed between the predicted probabilities and true labels". However, it is not clear whether this yields valid coverage. Since the resulting classifier $\hat{g}$ uses calibration data to learn the weights $\Theta$ the exchangeability between the calibration set and the test set is broken -- this is easy to see because information from the calibration set "leaks" into the weights $\Theta$. To see this differently: if you swap one calibration point with one test point the learned weights will be different. This is equivalent to why we cannot use one dataset to both train and calibrate the base classifier $f$ and why we need to either use Split CP, or use the full conformal approach. To maintain validity 3 datasets are necessary under the split framework: one for training, one for learning the augmentation policy, and one for the final calibration. However, this comes at a trade-off were we have to use smaller sets which is the same trade-off that standard split CP suffers from.

Note, the validity of the average variant is correct since here the exchangeability is maintained.

Note also that the fact that the emprical coverage matches the nominal coverage is not a proof.

The statement "We learn these weights using the calibration set by learning a set of weights that maximizes classifier accuracy by minimizing the cross-entropy loss computed between the predicted probabilities and true labels" shows the second weakness. The weights are trained to maximize the accuracy. However, this is not necessarily alligned with the actual goal of CP. It has been shown that we obtain the smallest prediction sets when the predicted probabilities (for all classes) match the ground-truth oracle probabilities (see e.g. [1] or APS).  However, the cross-entropy loss leads to over-confidence for the true class and does not encourage that the rest of the probabilities are well calibrated. Using a different loss such as e.g.  the one proposed by Stutz et al. (2022) which is mentioned in the related work or [1] is likely to lead to further improvements. Given the limited technical contributions such further experimental analysis is warranted.

Given that the learned weights were anyways close to 0 as reported, using fixed weights is another solution that maintains validity.

[1] "Training Uncertainty-Aware Classifiers with Conformalized Deep Learning" by Einbinder et al.

**Questions:**

1. Can you provide a rigorous proof that using the calibration set for learning the augmentation weights mantains validity, or fix the experiments using 3 seprate sets as outlined in the weaknesses section?

---

> ### Author Response · Authors · 2023-11-21
>
> Thanks for providing a detailed, constructive review. We’re glad you appreciate the simplicity of the approach and the detailed experimentation. Test-time augmentation’s simplicity (both in implementation and intuition) is one of the key strengths we wanted to demonstrate in the context of conformal prediction.
>
> **On fixing the experiments to use 3 separate sets of examples**: Excellent point; thank you for bringing it up. We have replicated all experiments by using three sets (the original training set, 20% of original calibration set for TTA, 80% of the original calibration set for identifying the threshold). We found that splitting the data has **no significant impact on the reported results**. We have modified the methods section to include this data split as a requirement, and have described the trade-off this incurs. We have also included additional experiments that measure the tradeoff between data used to learn the test-time augmentation policy and data used to identify the threshold (Figure 8); in short, the benefits of TTA are robust to any particular split, which suggests that the benefits TTA confers outweigh the cost to identifying the correct threshold.
>
> **On using a different loss to learn the TTA policy**: Yes, it could be that there are more appropriate losses to use to train the test-time augmentation policy. To study this, we ran additional experiments training the TTA policy using the focal loss and using the conformal training loss proposed by [1]. We found no significant improvements to the prediction set size when compared to the cross-entropy loss (Table 3 in the revised manuscript, reproduced below).
>
>
> | Dataset   |   alpha | Method                     | Average Prediction Set Size   |
> |:----------|--------:|:---------------------------|:------------------------------|
> | CUB-Birds |    0.01 | RAPS+TTA-Learned+CE        | 14.045 (1.323)                |
> | CUB-Birds |    0.01 | RAPS+TTA-Learned+Conformal | 13.776 (2.198)                |
> | CUB-Birds |    0.01 | RAPS+TTA-Learned+Focal     | 13.416 (1.991)                |
> | CUB-Birds |    0.05 | RAPS+TTA-Learned+CE        | 3.571 (0.576)                 |
> | CUB-Birds |    0.05 | RAPS+TTA-Learned+Conformal | 3.302 (0.312)                 |
> | CUB-Birds |    0.05 | RAPS+TTA-Learned+Focal     | 3.194 (0.202)                 |
> | CUB-Birds |    0.1  | RAPS+TTA-Learned+CE        | 1.893 (0.187)                 |
> | CUB-Birds |    0.1  | RAPS+TTA-Learned+Conformal | 1.865 (0.163)                 |
> | CUB-Birds |    0.1  | RAPS+TTA-Learned+Focal     | 1.791 (0.102)                 |
> | ImageNet  |    0.01 | RAPS+TTA-Learned+CE        | 31.248 (2.177)                |
> | ImageNet  |    0.01 | RAPS+TTA-Learned+Conformal | 32.257 (3.608)                |
> | ImageNet  |    0.01 | RAPS+TTA-Learned+Focal     | 32.612 (3.799)                |
> | ImageNet  |    0.05 | RAPS+TTA-Learned+CE        | 4.889 (0.168)                 |
> | ImageNet  |    0.05 | RAPS+TTA-Learned+Conformal | 4.867 (0.122)                 |
> | ImageNet  |    0.05 | RAPS+TTA-Learned+Focal     | 4.906 (0.195)                 |
> | ImageNet  |    0.1  | RAPS+TTA-Learned+CE        | 2.312 (0.054)                 |
> | ImageNet  |    0.1  | RAPS+TTA-Learned+Conformal | 2.308 (0.068)                 |
> | ImageNet  |    0.1  | RAPS+TTA-Learned+Focal     | 2.363 (0.085)                 |
>
>
> **On producing a rigorous proof that learning the weights could maintain validity**: we will think further on this. It is an exciting suggestion that we will consider beyond the rebuttal period.

---

> > ### Comment · Reviewer_aF3F · 2023-11-21
> > **Reply.**
> >
> > Thank for including the fix. I will raise my score.

---

### Official Review · Reviewer_BaaG · 2023-11-03

**Soundness:** 2 fair
**Presentation:** 3 good
**Contribution:** 2 fair
**Rating:** 6
**Confidence:** 2

**Summary:**

This paper adresses the problem of producing large prediction sets commonly seen in current approaches of conformal prediction, by applying test-time augmentation to the computation of conformal scores. To compute the conformal score of a sample, the proposed method uses a linear combination of estimated probabilities given by the base classifier on the original data vector and its augmentations. The weights of this linear combination are either uniform or learned by minimizing the cross entropy loss on the calibration set, giving rise to two algorithmes correspondingly referred as TTA-Avg and TTA-Learned.

An extensive empirical study was carried out to show the effectiveness of test-time augmentation in reducing the prediction set size. The superiority of TTA-Learned over TTA-Avg is notably observed for the expanded augmentation policy where some augmentations not used in the training step are introduced in the calibration step.

**Strengths:**

- This paper is well organized and easy to follow.

- Empirical evidence is provided to demonstrate the efficiency of the proposed method in reducing the size of the prediction set.

- A comprehensive empirical discussion is given to shed light on the behavior of the proposed method, and to explain intuitively its efficiency.

**Weaknesses:**

- Some of the technical claims might need further explanation (see Questions).

- The efficiency of the proposed method depends heavily on the applicability of test augmentation.

**Questions:**

- My biggest question about this work is on the assumption of data exchangeability in TTA-Learned. As the weights of augmentations in TTA-Learned are obtained by minimizing the cross entropy loss on the *calibration set*, there is a statistical dependence between the data in the calibration set and the learned weights of augmentations that are used to compute the conformal score. This means that calibration data and unseen exemples are not exchangeable with respect to the computation of conformal score. So I do not see how the assumption of exchangeability is preserved. If this point could be clarified, I would be willing to reconsider my score.


- As pointed out by the authors, it is understandable that TTA-Learned works better than TTA-Avg for the expanded augmentation policy as it allows the adjustment of the weights associated to the augmentations not included in the training. Meanwhile the results using APS reported in Table~4 show a close match between TTA-Learned and TTA-Avg. Could the authors provide some intuition behind that?

---

> ### Author Response · Authors · 2023-11-21
>
> Thanks for taking the time to provide a detailed review. We’re glad you found the experiments extensive, the presentation well-organized, and the explanations intuitive. We also appreciate your raising questions that should be addressed.
>
> **On using distinct examples to train the TTA policy**: Good point;  if the learned TTA policy is overfit to the calibration set, conformal scores computed on the calibration set will differ from conformal scores computed on the test set. This difference is practically important in the context of small calibration sets, which is an important distinction to make. **We reran all experiments by training the TTA policy on 20% of the calibration set and identifying the threshold based on the remaining 80% of examples and found largely no meaningful change in results**. We have updated the methods section to reflect this change.
>
> **On why TTA-Learned does not outperform TTA-Avg when combined with APS**: Although TTA-Learned improves Top-K accuracy compared to TTA-Avg (Figure 10), this occurs at K too low to affect the performance of APS. As earlier work has described [1], APS produces large prediction sets because of noisy estimates of small probabilities, which are then included in the prediction sets to meet the conformal threshold. Both TTA-Learned and TTA-Avg smooth the probabilities: they reduce the number of low-probability classes by aggregating predictions over perturbations of the image. The benefit that both TTA-Learned and TTA-Avg add to APS is thus similar to how RAPS penalizes classes with low probabilities. We have added this discussion to the revised manuscript.
>
> [1] Angelopoulos, Anastasios Nikolas, et al. "Uncertainty Sets for Image Classifiers using Conformal Prediction." International Conference on Learning Representations. 2020.

---

> > ### Comment · Reviewer_BaaG · 2023-11-22
> > **Reply to the authors**
> >
> > I thank the authors for their response and the proposed modifications. As stated in my review, my main concern is about the technical correctness of the conformal guarantee for TTA-Learned. This issue is fixed in the updated version by using two disjoint sets for learning the augmentation weights and calibration. My score is therefore raised.

---

### Author Response · Authors · 2023-11-21

Thanks to all for reading our work closely and providing constructive comments. We were pleased to hear that you found the experiments extensive, the presentation clear, and the idea innovative.  We have made a number of changes in response to your feedback and believe the resulting paper to be much stronger.

The new manuscript reflects these changes:

- Reviewers BaaG, aF3F, Gekq, and KHly raised concerns about reusing data for learning the TTA policy and choosing the conformal threshold. **We have updated all experiments to split the available calibration set into two disjoint subsets**: 20% of the examples are used to train the TTA policy and remainder are used to learn the calibration threshold. All other methods learn the threshold using the *entire* calibration set. **Interestingly, all results remain largely unchanged**. This suggests that the benefits TTA confers outweigh the cost of reducing the calibration set size for conformal predictors, a practically useful insight to those wishing to apply CP. We would not have discovered this without the reviewers’ feedback. **We also provide experiments where we vary the percentage of data used to train the test-time augmentation policy (Figure 10)**, and find that the results are not sensitive to this choice.
- Reviewers aF3F, Gekq, and L9GB suggested learning the test-time augmentation policy with an objective more closely aligned with smaller set size. **We provided additional experiments in which the test-time augmentation policy minimizes two alternate losses tied directly to efficiency in conformal predictors: the focal loss [1] and a conformal training loss [2]**. We provide results in Table 3. In short, **neither significantly outperforms the use of cross-entropy loss.**
- Reviewer L9GB suggested we include a table describing improvements in top-k accuracy; we have added this to the manuscript (Figure 10), demonstrating significant improvements in the optimal K required to achieve a pre-specified coverage on both ImageNet and iNaturalist.

[1] Lin, Tsung-Yi, et al. "Focal loss for dense object detection." Proceedings of the IEEE international conference on computer vision. 2017.

[2] Einbinder, Bat-Sheva, et al. "Training uncertainty-aware classifiers with conformalized deep learning." Advances in Neural Information Processing Systems 35 (2022): 22380-22395.

---

### Meta-Review · Area_Chair_U6rv · 2023-12-11

**Metareview:**

The paper introduce methodologies for reducing the size of the conformal prediction sets. The authors argue that the size is correlated with the quality of the score (which implicitly depends on the accuracy of predictors) and propose a data augmentation approach at test time. This leads to up to 30% improvement as demonstrated in the numerical experiments.

The reduction of size of the conformal set is quite nice and could be an important contribution if the method is proven to be valid. The latter is critically missing.


Conformal prediction provides framework to obtain valid prediction sets without assumption of the distribution. However, this distribution-free property does not holds for the *size*. Typically one needs to assume consistency or any control between the predictor being used and a ground-truth oracle predictors. As such it is quite difficult, perhaps impossible to get any strong claims on the size of the conformal sets without assumption on the accuracy of the predictors.

Also, the exchageanbility assumption seems violated by the data augmentations. Several reviewers explicitly mentioned this issue and I didn't find the authors reply acceptable. Conformal prediction is primarily about theoretically, provably, *valid* prediction sets. If this is not maintained, I believe that one can hardly argue about the significance of this paper in the CP community.

**Justification For Why Not Higher Score:**

Despite the encouraging numerical experiments, the soundness of the proposed method should be improved.
As a minimal requirement, the authors should explain (with a formal proof) why the validity of conformal prediction is maintained under this data augmentation setting.

**Justification For Why Not Lower Score:**

N/A

---

### Decision · Program_Chairs · 2024-01-16

Reject